# One Shot Inverse Reinforcement Learning for Stochastic Linear Bandits

**Etash Guha**[1,2]   **Jim James**[1]   **Krishna Acharya**[3]   **Vidya Muthukumar**[3,4]   **Ashwin Pananjady**[3,4]

[1]School of Computer Science, Georgia Tech, USA
[2]SambaNova Systems, Palo Alto, USA
[3]School of Industrial and Systems Engineering, Georgia Tech, USA
[4]School of Electrical and Computer Engineering, Georgia Tech, USA

## Abstract

The paradigm of inverse reinforcement learning (IRL) is used to specify the reward function of an agent purely from its actions and is critical for value alignment and AI safety. While IRL is successful in practice, theoretical guarantees remain nascent. Motivated by the need for IRL in large action spaces with limited data, we consider as a first step the problem of learning from a single sequence of actions (i.e., a demonstration) of a stochastic linear bandit algorithm. When the demonstrator employs the Phased Elimination algorithm, we develop a simple inverse learning procedure that estimates the linear reward function consistently in the time horizon with just a *single* demonstration. In particular, we show that our inverse learner approximates the true reward parameter within a error of $\mathcal{O}(T^{-\frac{\omega-1}{2\omega}})$ (where $T$ is the length of the demonstrator's trajectory and $\omega$ is a constant that depends on the geometry of the action set). We complement this result with an information-theoretic lower bound for any inverse learning procedure. We corroborate our theoretical results with simulations on synthetic data and a demonstration constructed from the MovieLens dataset.

## 1 INTRODUCTION

Using data-driven learning algorithms to design agents that interact with their environment has achieved great success in various fields ranging from robotics and video game playing to language models. As we deploy these learning algorithms and build machine-learning systems, it is important to ensure that they align with the goals of the human designer [Amodei et al., 2016], i.e., to understand how the human's reward is specified. However, alignment with designers' goals using hand-specified rewards is diffi-

cult and often mispecified [Anderson, 2001, MacGlashan and Littman, 2015]. Inverse Reinforcement Learning(IRL) [Abbeel and Ng, 2004, Ho and Ermon, 2016, Gershman, 2016, Fu et al., 2017, Jacq et al., 2019, Geng et al., 2020] is a well-established paradigm that circumvents the need for explicit reward specification and instead infers a reward function from demonstrations. In IRL, an inverse learner *only* observes the actions of a learned agent and then estimates the environment's reward function. The traditional IRL paradigm assumes that a demonstration consists of a roll-out of the optimal policy [Ng et al., 2000, Abbeel and Ng, 2004] or randomized variants [Ziebart et al., 2008, Ramachandran and Amir, 2007]. This paradigm has several limitations, including an often poor sample complexity—in particular, it requires multiple demonstrations. More crucially, even under simple scenarios (tabular RL/bandits), relying purely on demonstrations of an optimal policy can lead to a fundamental *identifability issue*; that is, more than one reward function explaining the demonstrator's actions. Such identifiability issues have been known since the early literature on IRL [Ng et al., 2000, Abbeel and Ng, 2004] and persist even with infinitely many demonstrations.

The *inverse bandit paradigm*, introduced by Guo et al. [2021], resolves both reward identifiability and sample complexity issues, albeit in the much simpler setting of stochastic multi-armed bandits (MAB). They show that it is possible to accurately estimate the reward structure by observing a *single online demonstration* of a low-regret bandit algorithm. In particular, they observe the demonstrator's behavior (i.e., the sequence of arms that it picks) *en route* to optimality and critically utilize the temporal information in online bandit learning to circumvent identifiability issues and the requirement of multiple demonstrations.

The question that motivates this paper is whether learning from a single demonstration in a similar manner is possible for more complex decision-making scenarios. In particular, we are interested in estimating the reward structure in the stochastic linear bandit setting by observing a single demonstration from a low-regret algorithm. This setting in itself is

*Accepted for the 40th Conference on Uncertainty in Artificial Intelligence* (UAI 2024).

much more challenging — the ideas from Guo et al. [2021] critically utilize the independence of reward distributions across arms in the MAB setting in multiple steps of the algorithm design and analysis and do not generalize to even the linear bandit case, which has highly structured rewards across actions.

In this paper, we show that it is indeed possible to estimate the linear reward parameter consistently in the time horizon from a single demonstration of the *phased elimination*[1] algorithm [Lattimore and Szepesvári, 2020]. To do so, we construct a simple inverse learning algorithm that uses an entirely different idea from the one in Guo et al. [2021]. Our algorithm selectively picks a small set of actions from the last epoch of the phased elimination demonstrator and forms a least squares estimate of the reward parameter. The actions are carefully selected to guarantee consistent estimation in the time horizon. Concretely, given an assumption on the density and "smoothness" of the action set (see Assumption 4.1), we show that our inverse learner with a *single* demonstration of length $T$ can estimate the reward function within an error of $T^{-\left(\frac{\omega-1}{2\omega}\right)}$, where $\omega \in [1, \infty)$ is a constant dependent on the smoothness of the action set. We also provide examples of action sets for which these assumptions are reasonable. In addition to the theory, we demonstrate the accuracy of our inverse learner on synthetic as well as semi-synthetic data.

**Contributions**    Our main contributions are listed in more detail below. Recall that the mean reward of an arm $a \in \mathbb{R}^d$ in the $d$-dimensional stochastic linear bandit setting is given by $\langle a, \theta^* \rangle$.

- We develop an inverse estimator of the reward parameter $\theta^*$ for a stochastic linear bandit instance from a single demonstration of the phased elimination algorithm. Our estimator consists of a least-squares estimate using: a) $d$ carefully selected arms from the last phase of elimination as covariates, and b) estimates of the rewards of these arms as responses. In Theorem 4.1, we prove an upper bound in the estimation error on the order of $\mathcal{O}(T^{-\frac{\omega-1}{2\omega}})$ where $T$ is the time horizon of the forward algorithm and $\omega \in [1, \infty)$ is a "smoothness" constant depending on the action set (see Assumption 4.1).

- In Theorem 5.1, we prove an information-theoretic lower bound of $\Omega\left(\sqrt{\frac{d}{T}}\right)$ on the optimal inverse estimator estimation error. When combined with our upper bound, this shows that as the action set gets "smoother" around the optimal arm, i.e. $\omega \to \infty$, our inverse estimator becomes information-theoretically optimal in its dependence on horizon $T$.

---

[1]Note that this is a natural generalization of successive-arm-elimination [Even-Dar et al., 2006] to linear bandits.

- We empirically evaluate our inverse learning algorithm on synthetic and semi-synthetic data, performing simulations on commonly used action sets such as the $\ell_1, \ell_2$, and $\ell_5$ ball. We then consider an application involving linear bandit algorithms for a recommender system on the MovieLens data set [Zhu and Kveton, 2022]. In particular, we model the problem of predicting the user's "preference vector" as an inverse linear stochastic bandit problem. We demonstrate that our inverse algorithm can efficiently predict the reward parameter of a user by observing the movies chosen by the recommender system. This could have downstream relevance in predicting the user's preference for movies not seen by the recommender system.

**Outline of paper**    We first provide a brief discussion of the most closely related work in Section 2, and then provide basic background for the stochastic linear bandit problem and phased elimination in Section 3. Section 4 discusses the methodology and proof outline of our main results, Section 5 states the information-theoretic lower bound, and we present our experiments in Section 6. We conclude with a discussion and future work in Section 7.

## 2    RELATED WORK

We organize our related work along two verticals: low-regret algorithms for stochastic linear bandits—which we call *forward algorithms* in our setting—and inverse algorithms for reinforcement learning.

### 2.1    STOCHASTIC LINEAR BANDITS

The setting of stochastic linear bandits was first analyzed by Abe and Long [1999]; since then, several algorithms have been proposed that achieve a regret bound of $\mathcal{O}(d\sqrt{T})$ for infinite action sets, and $\widetilde{\mathcal{O}}(\sqrt{d \log K T})$ for action sets of size $K$, e.g. [Dani et al., 2008, Chu et al., 2011, Abbasi-Yadkori et al., 2011, Valko et al., 2014]. In both cases, these upper bounds are matched by information-theoretic lower bounds [Lattimore and Szepesvári, 2020]. In this paper, we assume that the demonstrator is the Phased Elimination algorithm in Lattimore and Szepesvári [2020], Valko et al. [2014], Esfandiari et al. [2019], which also achieves the optimal $\mathcal{O}(\sqrt{d \log K T})$ regret bound for stochastic linear bandits with a finite action set. This algorithm is related to the successive-arm-elimination (SAE) algorithm [Even-Dar et al., 2006], which was shown to be compatible with inverse learning in the MAB setting [Guo et al., 2021]. However, the phased-elimination algorithm has key differences, including the non-uniform sampling scheme among active arms in each epoch and a doubling in epoch length in each increment. The doubling of epochs, which is not part of SAE for MAB, turns out to be particularly challenging to deal

with in inverse estimation for linear bandits. At the same time, the doubling trick is essential for the algorithm itself to attain sublinear regret in the stochastic linear bandit.

## 2.2 INVERSE REINFORCEMENT LEARNING

The original works on IRL [Ng et al., 2000, Abbeel and Ng, 2004] noted an identifiability issue in the reward function from an optimal demonstration that cannot be resolved except in special cases involving additional structure on the reward or additional side information [Gershman, 2016, Amin et al., 2017, Fu et al., 2017, Geng et al., 2020]. Assuming randomized variants of the optimal policy (e.g. max-entropy IRL [Ziebart et al., 2008], Bayesian IRL [Ramachandran and Amir, 2007]) can partially alleviate this identifiability issue, but only in special cases. The identifiability issue remains open for the inverse problem in RL, but was resolved in Guo et al. [2021] for stochastic MAB by considering a single exploring demonstrator. Aside from this inverse bandit paradigm, the works of Gao et al. [2018] and Jacq et al. [2019] introduced a related paradigm of "learning from learners", but used optimization instead of bandit learning for the demonstration and still require several demonstrations. More recently, Hüyük et al. [2022] considered one-shot inverse learning from a single demonstration of a certain type of Bayesian *contextual bandit* algorithm. Their algorithms are based on approximate Bayesian inference and are empirically successful, but do not come with a guarantee of consistency. Finally, we note that there are distinct objectives for learning from demonstrations that can be far easier than IRL; for example, imitation learning [Ho and Ermon, 2016] or apprenticeship learning [Abbeel and Ng, 2004, Shani et al., 2022]. These tasks usually do not suffer from the same identifiability issues as IRL.

## 3 PROBLEM FORMULATION

We now discuss the basic setup for the inverse linear bandit problem. In Section 3.1, we discuss preliminaries for the stochastic linear bandit problem, and in Section 3.2, we describe the forward algorithm that we assume the demonstrator will use, i.e. the phased-elimination algorithm [Lattimore and Szepesvári, 2020, Valko et al., 2014]. We then formalize the inverse linear bandit problem and our desired estimation error guarantee in Section 3.3.

### 3.1 PRELIMINARIES ON STOCHASTIC BANDITS

Our environment is defined as a structured, parameterized bandit instance $\mathcal{M} = (\theta^*, \mathcal{A})$, where $\theta^*$ parameterizes the reward function of the environment and $\mathcal{A}$ is a finite (but potentially large) set of actions the forward algorithm may take while interacting with the environment. A *forward algorithm* sequentially interacts with this environment over

$T$ rounds. At round $t$, the algorithm chooses an action from the action set[2], $a_t \in \mathcal{A}$ and receives a reward given by

$$x_t := G_{\theta^*}(a_t) + \eta_t,$$

where $G_{\theta^*}(a)$ is the mean reward function parameterized by $\theta^*$ and $\eta_t$ denotes noise, which we assume to be zero-mean and 1-sub-Gaussian. The forward algorithm repeats this procedure for $T$ steps. The main property that we desire from the forward algorithm is to minimize *pseudo-regret*, defined as

$$R_T = \sum_{t=1}^{T} \max_{a \in \mathcal{A}} G_{\theta^*}(a) - G_{\theta^*}(a_t).$$

As is standard in the bandit literature [Lattimore and Szepesvári, 2020], we desire in particular that $R_T = \widetilde{o}(T)$, i.e. sublinear regret in the total number of rounds $T$. We consider the special case of the stochastic linear bandit for this work. Here, $\mathcal{A} \subset \mathbb{R}^d$ and $\theta^* \in \mathbb{R}^d$, and the mean reward function is defined as $G_{\theta^*}(a) = \langle a, \theta^* \rangle$.

### 3.2 THE FORWARD ALGORITHM: PHASED ELIMINATION

Inspired by the relative simplicity of the inverse error analysis of the *successive-arm-elimination* algorithm [Even-Dar et al., 2006] for stochastic multi-armed bandits presented in Guo et al. [2021], we will assume that the forward algorithm uses its natural counterpart for the linear bandit problem, which is commonly called *phased elimination* [Lattimore and Szepesvári, 2020, Valko et al., 2014]. While not as popular in practice as LinUCB [Abbasi-Yadkori et al., 2011] and linear Thompson sampling [Agrawal and Goyal, 2013], the phased elimination satisfies a similar (optimal) sublinear regret guarantee, given by $R_T = \widetilde{\mathcal{O}}(\sqrt{dT \log |\mathcal{A}|})$. It has found particular use in bandit instances on smooth functions on a graph [Valko et al., 2014].

To keep the paper self-contained, we recap the salient properties of the phased elimination algorithm, which we also formally define in Algorithm 1. At a high level, the algorithm operates in phases that increase in length and eliminates a subset of arms at the end of each phase. Consider a phase $\ell \geq 1$, and denote the set of active arms at the beginning of phase $\ell$ by $\mathcal{A}_\ell$. The algorithm first solves a convex optimization problem to pick a *G-optimal design* $\{\pi(a)\}_{a \in \mathcal{A}_\ell}$; see [Lattimore and Szepesvári, 2020].

**Definition 3.1.** *A* G-optimal design *for an action set $\mathcal{A}$ at phase $\ell \geq 1$ is a function $\pi_\ell : \mathcal{A} \to \mathbb{R}_+$ that maximizes $f(\pi) = \log(\det(V(\pi)))$ such that $\sum_{a \in \mathcal{A}} \pi(a) = 1$, where $V(\pi) = \sum_{a \in \mathcal{A}} n_\ell(a) a a^T$ and $n_\ell(a) =$*

---

[2]Note that the algorithm has access to the prior history $\{a_1, x_1, a_2, x_2, \ldots, a_{t-1}, x_{t-1}\}$ and can use this history as input to decide an action $a_t$ at round $t$.

**Algorithm 1:** Phased Elimination
___
**Input :** $\delta$ (probability parameters), $T$ (total number of rounds), $\{\nu_1, \ldots, \nu_L\}$ (error parameters)

**Result:** $a_1, \ldots, a_T$

1   $\ell \leftarrow 0$

2   $\mathcal{A}_1 \leftarrow \mathcal{A}$

3   **while** *Number of rounds $\leq T$* **do**

4     $\varepsilon_\ell \leftarrow 2^{-\ell}$

5     $\pi_\ell \leftarrow$
     G-Optimal design of $\mathcal{A}_\ell$ as a function of $\delta$ and $\nu_\ell$

6     $N_\ell \leftarrow 0$

7     Play each action $a \in \mathcal{A}_\ell$ each $n_\ell(a) =$
     $\left\lceil \frac{2d\pi_\ell(a)}{\nu_\ell^2} \log\left(\frac{|\mathcal{A}|\ell(\ell+1)}{\delta}\right) \right\rceil$ times

8     $V_\ell \leftarrow \sum_{a \in \mathcal{A}_\ell} n_\ell(a) aa^T$

9     $\theta_\ell \leftarrow V_\ell^{-1} \sum_{t=t_\ell}^{t_\ell + T_\ell} a_t x_t$

10    $\mathcal{A}_{\ell+1} \leftarrow \{a \in \mathcal{A}_\ell \text{ s.t. } \max_{b \in \mathcal{A}_\ell}\langle\theta_l, b-a\rangle \leq 2\varepsilon_l\}$

11    $\ell \leftarrow \ell + 1$

12 **end**
___

$\left\lceil \frac{2d\pi_\ell(a)}{\nu_\ell^2} \log\left(\frac{|\mathcal{A}|\ell(\ell+1)}{\delta}\right) \right\rceil$. *Note that $\nu_\ell$ and $\delta > 0$ are input parameters to the G-optimal design algorithm.*

After solving for $\pi_\ell$, the algorithm pulls $a \in \mathcal{A}$ exactly $\left\lceil \frac{2d\pi_\ell(a)}{\nu_\ell^2} \log\left(\frac{|\mathcal{A}|\ell(\ell+1)}{\delta}\right) \right\rceil$ times, where $\delta$ denotes the allowed probability of failure and $\nu_\ell$ is a error parameter. At the end of phase $\ell$, the algorithm uses the observed rewards in phase $\ell$ alone to construct a least-squares estimate of the reward parameter, denoted by $\theta_\ell$. It then eliminates all arms that are suboptimal below a confidence width given by the structure of the linear model (see Lemma A.1). As long as $\nu_\ell \leq \epsilon_\ell := 2^{-\ell}$, this algorithm is known to achieve the optimal regret bound $R_T = \mathcal{O}\left(\sqrt{dT \log\left(\frac{|\mathcal{A}| \log(T))}{\delta}\right)}\right)$ for finite action sets [Lattimore and Szepesvári, 2020].

### 3.3 THE INVERSE LINEAR BANDIT PROBLEM

We now define the inverse linear bandit problem. The inverse learner is assumed to have access to the sequence of actions $(a_1, \ldots, a_T)$ and the action sets at each phase $(\mathcal{A}_1, \ldots, \mathcal{A}_L)$ from a *single demonstration* of the phased elimination algorithm defined in Section 3.2. Importantly, the learner *cannot* access the corresponding sequence of rewards $(x_1, \ldots, x_T)$. As in Guo et al. [2021], we also assume access[3] to the best reward $\mu^* = \max_{a \in \mathcal{A}}\langle a, \theta^*\rangle$ as well as the optimal arm $a^* = \arg\max_{a \in \mathcal{A}}\langle a, \theta^*\rangle$. Our goal is to construct an estimate $\hat{\theta}$ with small relative error to the true

___
[3]As in Guo et al. [2021], one can relax these assumption if we restrict ourselves to estimating rewards up to additive shift of $\mu^*$, and use a near-optimal, most frequently pulled arm instead of $a^*$.

reward parameter $\theta^*$, defined as $\frac{\|\hat{\theta}-\theta^*\|_2}{\|\theta^*\|_2}$. We also make the following assumptions on the forward algorithm.

**Assumption 3.1.** *[Assumptions on forward algorithm]*

1. *The total number of phases $L$ executed by our forward algorithm is upper bounded by $\bar{L} \in \mathbb{N}$.*

2. *The error parameter at each phase $\nu_\ell = \iota\epsilon_\ell$ is chosen such that $0 < \iota < 1$.*

## 4 METHODOLOGY AND MAIN RESULT

In this section, we describe the methodology for our inverse learning approach. We first define some notation specific to this section. We will define the two-dimensional subspace spanned by two vectors $u$ and $v$ as span$(u, v)$. For a set of vectors $\mathcal{C} = \{c_1, c_2, \ldots, c_n\}$, we define its condition number as cond$(\mathcal{C}) = $ cond $\left(\begin{bmatrix} c_1 & c_2 & \ldots & c_n\end{bmatrix}^\top\right)$, where the vectors constitute the rows of the matrix.

The goal of the inverse learner is to learn the environment's true reward parameter $\theta^* \in \mathbb{R}^d$. As mentioned in the introduction, a core challenge in the linear bandit setting is the shared structure across arms — pulling an arm $a$ will change the forward algorithm's estimates of all arms $a' \neq a$, rendering an out-of-the-box approach from Guo et al. [2021] infeasible. At the same time, this shared structure could help estimate $\theta^*$ if one could reliably estimate the rewards of a large and "well-behaved" subset of actions. To be more concrete, suppose that we had an oracle where for any arm $a$ in some well-conditioned set $\mathcal{A}^e \subset \mathcal{A}$, we knew its exact mean reward $G_{\theta^*}(a)$. In this case, the optimal estimator of $\theta^*$ would minimize the least-squared error between the rewards and the arms, i.e.

$$\hat{\theta} = \arg\min \sum_{a \in \mathcal{A}^e} \left(G_{\theta^*}(a) - \langle a, \hat{\theta}\rangle\right)^2. \quad (1)$$

With this intuition, our inverse learner (Algorithm 2) proceeds in three steps: a) constructing a specific action subset $\mathcal{A}^e$ (Steps 4-6 of Algorithm 2), b) estimating the reward $G_{\theta^*}(a)$ for each $a \in \mathcal{A}^e$ (Step 8 of Algorithm 2), and c) computing the least squares estimate of $\theta^*$ using the reward estimates from step (b) (Step 8 of Algorithm 2). Note that the demonstrator chooses $\delta$ and $\iota$, which are inputs to the algorithm. Equation (1) suggests that, with near-perfect access to mean rewards, one might want to select the subset $\mathcal{A}^e$ to be as large as possible. However, this is misleading reasoning for several reasons: first, different arms are pulled an unequal number of times due to elimination, meaning that the mean rewards of certain arms can be estimated more reliably than others; second, selecting arms that are too close to each other (i.e., arms $a, a'$ for which $\|a - a'\|_2$ is too small) would result in the estimation error blowing up due to poor conditioning of the action set $\mathcal{A}^e$.

The crux of our methodology lies in carefully designing the action subset $\mathcal{A}^e$ to adequately control the estimation error that arises due to the finite sample regime as well as the condition number of the design matrix in Equation (1). We now describe each of step in Algorithm 2 in more detail.

## 4.1 CONSTRUCTION OF ACTION SUBSET $\mathcal{A}^e$

---
**Algorithm 2:** Inverse Estimator (also see equation 2)

---
**Data:** Set of active arms at each epoch$(\mathcal{A}_1, \ldots, \mathcal{A}_L)$
**Result:** Estimated reward parameter $\hat{\theta}$

1   $\mathcal{A}^e = \{\}$
2   $\beta := (3(1-\iota)\epsilon_L)^{\frac{1}{\omega}}$
3   **for** $i \in [d]$ **do**
4      **if** $\exists a \in \mathcal{A}$ s.t. $\tau(a, i) \geq \beta, \mathrm{dist}(a, i) \leq \gamma, a \in$
       $\mathcal{A}_L \setminus \mathcal{A}_{L-1}$ **then**
5          $\mathcal{A}^e \leftarrow \mathcal{A}^e \cup \{a\}$
6      **end**
7   **end**
8   $\hat{\theta} = \arg\min \sum_{a \in \mathcal{A}^e} \left( \mu^* - 2(1+\iota)\epsilon_L - \langle a, \hat{\theta} \rangle \right)^2$
9   **return** $\hat{\theta}$

---

In this section we describe the first part of the algorithm (Steps 4-6 in Algorithm 2), which constructs the action subset $\mathcal{A}^e$. We select arms only from the last eliminated set, i.e. $\mathcal{A}^e \subset (\mathcal{A}_L \setminus \mathcal{A}_{L-1})$, to ensure that the mean reward of each arm in $\mathcal{A}^d$ can be estimated as accurately as possible. We also aim to select arms with as large as possible pairwise angles between each other in order to appropriately control the condition number of the design matrix in Equation (1). We will pick $d$ arms in $d$ evenly spaced planes to ensure the latter property. In particular, we will select the $i$-th arm to be in the subspace spanned by the optimal arm $a^*$ and the $i$-th vertex of a $d-1$-regular simplex.

Formally, consider any $d-1$-regular simplex $\mathcal{S}_i$ in $\mathbb{R}^d$ formed by the unit vectors $\{s_1, \ldots, s_d\}$ such that $s_i \neq \alpha a^*$ for any $i \in [d], \alpha \in \mathbb{R}$. To form the $i$-th arm in $\mathcal{A}^e$, we will iterate through each arm $a$ in the action set $\mathcal{A}$ and calculate two relevant metrics. The first is the distance between an arm $a$ and its projection onto the subspace $\mathrm{span}(a^*, s_i)$. Formally, let $\mathrm{proj}(a, i)$ denote the vector obtained by projecting an arm $a \in \mathcal{A}$ to the two dimensional subspace $\mathrm{span}(a^*, s_i)$, i.e. $\mathrm{proj}(a, i) := \arg\min_{a' \in \mathrm{span}(a^*, s_i)} \|a - a'\|_2$. Then, define the distance between an arm $a$ and the plane $\mathrm{span}(a^*, s_i)$, as

$$\mathrm{dist}(a, i) := \|\mathrm{proj}(a, i) - a\|_2. \tag{2a}$$

The second metric we will calculate is the angle formed between the projection $\mathrm{proj}(a, i)$ and the optimal arm $a^*$, which we will denote as

$$\tau(a, i) := \cos^{-1} \left( \frac{\langle \mathrm{proj}(a, i), a^* \rangle}{\|\mathrm{proj}(a, i)\|_2 \|a^*\|_2} \right). \tag{2b}$$

Our goal is to find a subset of $d$ arms $\mathcal{A}^e = \{a^1, \ldots, a^d\}$ such that for the $i$-th arm $a^i$, a) $\mathrm{dist}(a^i, i)$ is small, b) $\tau(a^i, i)$ is large (ensuring good conditioning of the action set), and c) $a^i \in \mathcal{A}_L \setminus \mathcal{A}_{L-1}$, i.e. $a^i$ was eliminated in phase $L$ (ensuring a reliable estimate of the mean reward of $a^i$).

It is worth noting that this specific subset of arms, $\mathcal{A}^e$, may not exist for an arbitrary action set $\mathcal{A}$ if the action set is not sufficiently dense or is very "sharp" around the optimal arm. Below, we state our assumptions on the action set to rule out these possibilities.

**Assumption 4.1.** *We assume that there exists a value $L$ such that for all $i \in [d]$, for all $\ell \in [L]$, and some $\omega > 1$, there exists an arm $a^i$ with the properties:*

1. $\tau(a^i, i) \geq \beta$ where $\beta := (3(1-\iota)\epsilon_L)^{\frac{1}{\omega}}$
2. $\mathrm{dist}(a^i, i) \leq \gamma \leq \frac{\epsilon_L}{\|\theta^*\|_2 d}$.
3. $\mu^* - 4(1-\iota)\epsilon_L \leq \langle \theta^*, a^* - a^i \rangle \leq \mu^* - 2(1-\iota)\epsilon_L$.

As articulated above, Part 1 of the assumption ensures that the angle between $\mathrm{proj}(a^i, i)$ and the optimal arm $a^*$ is sufficiently large; Part 2 ensures that $a^i$ is close to its projection onto the plane[4] given by $\mathrm{span}(a^*, i)$; and Part 3 ensures that the arm $a^i$ is sufficiently suboptimal to be eliminated in phase $L$ with high probability, but also sufficiently high in reward to stay active until phase $L$ with high probability:

**Lemma 4.1.** *Any arm $a$ close to the optimal arm satisfying*

$$2(1-\iota)\epsilon_\ell < \langle a^* - a, \theta^* \rangle \leq 4(1-\iota)\epsilon_\ell \tag{3}$$

*will be in $\mathcal{A}_\ell \setminus \mathcal{A}_{\ell-1}$ with probability at least $1 - |\mathcal{A}|L\delta$. Therefore, with probability at least $1 - |\mathcal{A}|L\delta$, the mean reward of any arm $a \in \mathcal{A}_\ell \setminus \mathcal{A}_{\ell-1}$ is bounded as*

$$\mu^* - 4(1-\iota)\epsilon_\ell \leq \langle a, \theta^* \rangle \leq \mu^*.$$

This statement is proved in Appendix A.

Note that one can find arms satisfying Part 2 of Assumption 4.1 as long as the action set is sufficiently "dense" (in the sense of satisfying a $\gamma$-covering of some continuous set[5] in $\mathbb{R}^d$), and it is easy to find arms satisfying Parts 1 and 3 as long as the action set is sufficiently "smooth" around $a^*$, meaning that arms with a reward bounded away from the optimal reward *and* with a sufficiently large angular distance from $a^*$ can be found. We comment further on natural action sets satisfying all of these assumptions in Section 4.4.

As long as $\mathcal{A}^e$ can be selected in this way, its condition number is upper bounded according to the following lemma.

---
[4]Note that this implicitly requires the action set to span $\mathbb{R}^d$.
[5]This is a natural setting since if the action set is continuous, then it is common to run the forward algorithm on a $\gamma$-covering.

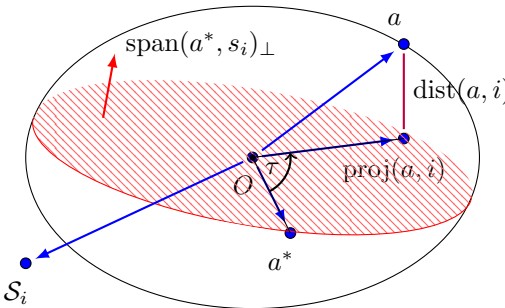

Figure 1: A visualization of the formation of an arm in $\mathcal{A}^e$. Here, we project an arm $a$ onto the subspace $\text{span}(a^*, s_i)$ such that $\tau(a, i)$, the angle between the projection and $a^*$, is large and $\text{dist}(a, i)$ is small.

**Lemma 4.2** (**Condition Number of $\mathcal{A}^e$**). *Let $\chi_2$ and $\chi_1$ be defined as $\chi_2 = \max_{a \in \mathcal{A}} \|a\|_2, \chi_1 = \min_{a \in \mathcal{A}} \|a\|_2$. Suppose that Assumption 4.1 holds, and we can select the action subset $\mathcal{A}^e$ according to Steps 4-6 of Algorithm 2. Then, with probability at least $1 - |\mathcal{A}|L\delta$, the condition number of the matrix whose rows are elements of $\mathcal{A}^e$ satisfies*

$$\text{cond}(\mathcal{A}^e) \leq \frac{\chi_2 + \gamma\sqrt{d}}{\chi_1 \left[ (2d)^{-\frac{1}{2}}\beta \right] - \gamma\sqrt{d}}.$$

This lemma is proved in Appendix B.

### 4.2 ESTIMATING THE REWARDS OF ACTIONS IN $\mathcal{A}^e$

We next estimate the mean reward for each of the arms from $\mathcal{A}^e$, i.e. $G_{\theta^*}(a) := \langle a, \theta^* \rangle$ for all $a \in \mathcal{A}^e$, and provide upper bounds on the estimation error of each of these rewards. Since each arm belongs to $\mathcal{A}_L \setminus \mathcal{A}_{L-1}$, it will have a mean reward less than the optimal reward $\mu^*$ and greater than $\mu^* - 4(1 + \iota)\epsilon_\ell$ from Lemma 4.1. Consequently, the simple estimate $\hat{r} := \mu^* - 2(1 + \iota)\epsilon_\ell$ satisfies the following upper bound on the estimation error.

**Lemma 4.3.** *Let $r$ denote the vector of true rewards $\left\{ R_{\theta^*}(a^i) \right\}_{i=1}^d$ and $\hat{r}$ denote a vector of our estimated rewards given by $\{\mu^* - 2(1 + \iota)\epsilon_\ell\}_{i=1}^d$. Then, we have $\frac{\|r - \hat{r}\|_2}{\|r\|_2} \leq \frac{4\epsilon_L}{\mu^* - 8\epsilon_L} = \mathcal{O}\left(2^{-L}\right)$ with probability at least $1 - |\mathcal{A}|L\delta$.*

This lemma is proved in Appendix B.

### 4.3 MAIN RESULT: ESTIMATION ERROR BOUND

The final step (Step 8 of Algorithm 2) constructs $\hat{\theta}$ as the least-squares estimate (Equation (1) using the action set $\mathcal{A}^e := \{a^1, \ldots, a^d\}$ as covariates and estimated rewards

$\{\hat{r}\}_{i=1}^d$ as responses. We now present our main result, which is the error guarantee of the estimator from Algorithm 2.

**Theorem 4.1.** *Let $\chi_2 = \max_{a \in \mathcal{A}} \|a\|_2$, $\chi_1 = \min_{a \in \mathcal{A}} \|a\|_2$ and define $J = \log\left( \frac{|\mathcal{A}|L(L+1)}{\delta} \right)$ as shorthand. Then, we have*

$$\frac{\left\| \hat{\theta} - \theta^* \right\|_2}{\|\theta^*\|_2} = \mathcal{O}\left( \frac{\chi_2 d^{\frac{2\omega - 1}{2\omega}} J^{\frac{\omega - 1}{\omega}}}{\chi_1 T^{\frac{\omega - 1}{2\omega}}} \right)$$

*with probability at least $1 - |\mathcal{A}|L\delta$. Note that $\omega > 1$ is the constant from Assumption 4.1.*

Theorem 4.1 is proved in Appendix B. Since we have assumed $\omega > 1$ in Assumption 4.1, Theorem 4.1 implies consistent estimation of $\theta^*$ as $T \to \infty$. Moreover, if the smoothness parameter $\omega \to \infty$, the dependence on $d$ becomes linear, and the dependence on $T$ becomes $T^{-1/2}$; the latter is optimal in its dependence on $T$ as shown in our forthcoming information-theoretic lower bound (Theorem 5.1).

### 4.4 DISCUSSION ON VIABILITY OF ASSUMPTIONS

A natural question is whether the assumptions made on the action set are reasonable and whether the value of $\omega$ can be characterized for arbitrary action sets. The lemma below is a proof-of-concept that for each $\omega \in [1, \infty)$ there exist a valid action set that satisfies Assumption 4.1. A quantitative version of this result is stated and proved in Appendix D.

**Lemma 4.4.** *Given any value $\omega \in [1, \infty)$, there exists a linear bandit instance (i.e., a set of arms and a linear reward function) that satisfies Assumption 4.1.*

Qualitatively, an example set that defines such a bandit instance exists even in two dimensions. One can construct it with the optimal arm is at point $(1, 0)$ and there exist two adjacent arms that are equiangular with the optimal arm while having sufficient magnitude to have a certain reward that is specified in the construction. We provide a sample visualization in Figure 4 of the appendix. In higher dimension, a natural tensorization of such an instance will satisfy the assumption.

## 5 INFORMATION-THEORETIC LOWER BOUND

We now provide an information-theoretic lower bound on the accuracy achievable by any inverse estimator via the classical Le Cam binary testing approach [LeCam, 1973]. Essentially, this approach creates two different bandit instances $\mathcal{M}_1 = (\theta_1^*, \mathcal{A}_1)$ and $\mathcal{M}_2 = (\theta_2^*, \mathcal{A}_2)$ and has the

forward algorithm work with one of these bandit instances. Then, we show that the inverse algorithm will be unable to distinguish which of the bandit instances the forward algorithm interacted with given a single demonstration of *any* forward algorithm that incurs regret at least $\widetilde{\mathcal{O}}(\sqrt{dT})$ and sufficiently explores each direction. Since the fundamental limit on regret for stochastic linear bandits for finite action sets is known to be $\widetilde{\mathcal{O}}(\sqrt{dT})$ [Lattimore and Szepesvári, 2020], this implies a fundamental limit on inverse estimation. Theorem 5.1 is proved in Appendix C.

**Theorem 5.1.** *For a bandit instance $\mathcal{M}$ characterized by reward parameter $\theta_1^*$ and action set $\mathcal{A}$, there exists a bandit instance $\mathcal{M}'$ with parameter $\theta_2^*$ and the same action set $\mathcal{A}$ such that any inverse estimator incurs error*

$$\max\{\|\hat{\theta} - \theta_2^*\|_2, \|\hat{\theta} - \theta_1^*\|_2\} = \widetilde{\Omega}\left(\sqrt{\frac{d}{T}}\right).$$

# 6  EXPERIMENTS

To validate our results empirically, we implement our inverse estimator on both simulated and semi-synthetic environments, measuring the error in the estimate of $\theta^*$. To run Phased Elimination and our estimator on these action sets most naturally, we run the algorithm for a fixed number of phases rather than a fixed number of rounds; see Appendix E for a formal description.

## 6.1  SIMULATIONS

To construct our action sets, we sample 4000 vectors from the surface of the unit $\ell_1$, $\ell_2$, and $\ell_5$ balls and use this finite set as $\mathcal{A}$. This is done by independently sampling each entry from a generalized Gaussian distribution (having density proportional to $\exp(-|x|^\beta)$) with a $\beta = 1, 2$, and 5 respectively, and then normalizing the resulting vector by its respective $\ell_p$ norm [Barthe et al., 2005]. The noise in the observed reward is Gaussian with mean 0 and variance 0.02.

Using the implementation in Algorithm 3, we run 100 trials of a bandit instance with maximum number of phases $L \in \{3, 4, 5, 6\}$ and dimensions $d \in \{3, \ldots, 8\}$. Afterward, we run the inverse estimator on each instance and measure the metric of relative error of $\hat{\theta}$, defined as $\frac{\|\hat{\theta} - \theta^*\|_2}{\|\theta^*\|_2}$. We record this error for the last round of the final phase.

On the one hand, from Theorem 4.1 we expect relative error to decay with the total number of rounds $T$. From the log-log plots in Figure 2, we observe that this trend holds for each action set by examining the trend of each best-fit line. The lines in a Figure 2(a), Figure 2(b), and Figure 2(c) each contain slopes in the range $[-0.487, -0.413]$, indicating a polynomial rate of decay in $T$. On the other hand, Theorem 4.1 also predicts that relative error should

increase in $d$. In Figure 3, we plot the relative error of our inverse estimator on each unit ball for each dimension from 3 to 8, verifying that higher dimensional action sets indeed incur higher relative error. Furthermore, from the results in Table 1, we observe that at dimensions of 6 or higher, the inverse algorithm performs comparably to the forward algorithm's estimate $\hat{\theta}$ from the final round, occasionally incurring less relative error.

## 6.2  SEMI-SYNTHETIC EXPERIMENTS

To validate the performance of our estimator with more realistic data, we simulate the task of recommending movies to users on the MovieLens 25M dataset Lam and Herlocker [2006], Harper and Konstan [2015], as well as recommending music to users based on the digital music reviews subset of the Amazon Reviews dataset Hou et al. [2024]. The MovieLens 25M dataset consists of 25 million ratings across 160,000 users and 60,000 movies, while the Amazon Reviews digital music dataset contains 101,000 users, 70,000 songs, and 130,000 ratings. We follow a similar set up as by Zhu and Kveton [2022]. To create an action set and $\theta^*$, we randomly sample $u = 6,000$ users, $m = 4,000$ items, and their corresponding ratings from each dataset. We then perform a matrix factorization on $R \in \mathbb{R}^{u \times m}$, the matrix of ratings for each user and item, using Alternating Least Squares. This yields matrices $U$ and $M$ where $UM^\top = R$, $U \in \mathbb{R}^{u \times d}$, and $M \in \mathbb{R}^{m \times d}$. Therefore, each row in $M$ is a $d$ dimensional embedding corresponding to a item, while each row in $U$ corresponds to the reward parameter for a given user. We then simulate a user's choices and ratings by randomly sampling a reward parameter $\theta^* = U_i$, and running Algorithm 3 with $M$ as the set of arms for 6 phases. Afterward, we estimate the user's reward parameter via Algorithm 2. We repeat this for ten randomly selected users and average the relative error of $\hat{\theta}$ to generate of the entries in Table 2 for a fixed dimension $d$. We also repeat the entire experiment for four different values of $d$. Our numerical results are summarized in Table 2. As before, both inverse and forward estimation error increase with the dimension of the action set.

# 7  DISCUSSION

We have presented an inverse reinforcement learning algorithm for the setting of linear stochastic bandits and guarantees its convergence behavior as a function of the length of the demonstrator's trajectory. We empirically verified the efficacy of our algorithm in both simulation and semi-synthetic settings. Moreover, we showed a lower bound on the best achievable error by any inverse learner. An interesting future direction would be to extend a similar framework to nonlinear reward functions and general bandit settings.

A fundamental limitation of our work, even in the linear

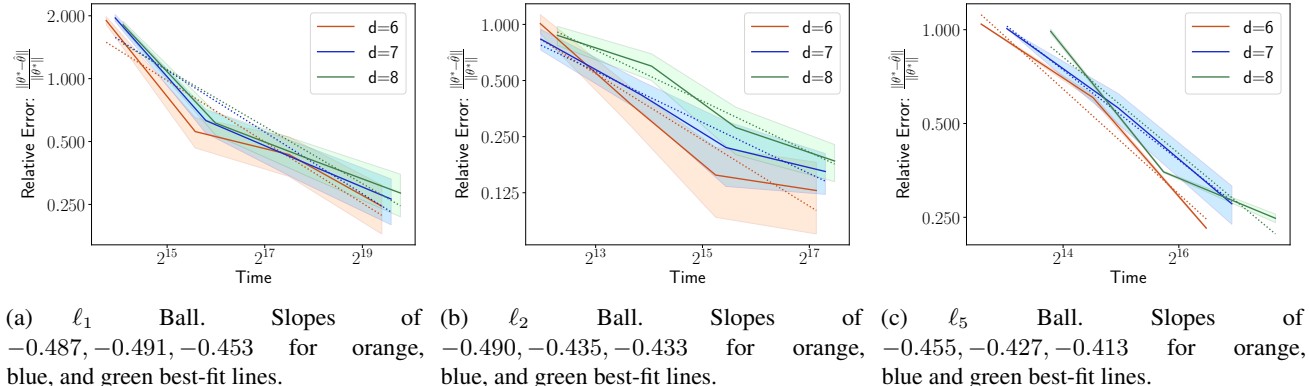

(a) $\ell_1$ Ball. Slopes of $-0.487, -0.491, -0.453$ for orange, blue, and green best-fit lines.

(b) $\ell_2$ Ball. Slopes of $-0.490, -0.435, -0.433$ for orange, blue, and green best-fit lines.

(c) $\ell_5$ Ball. Slopes of $-0.455, -0.427, -0.413$ for orange, blue and green best-fit lines.

Figure 2: The inverse estimator's performance (averaged over 100 trials) over the $\ell_1$, $\ell_2$, and $\ell_5$ balls across dimensions $d = 6, 7, 8$. The shaded region represents the standard deviation corresponding to each phase. Each graph is a log-log scale with orange, blue, and green dotted lines denoting a log-log linear fit for each dimension.

| | $\ell_1$ BALL | | $\ell_2$ BALL | | $\ell_5$ BALL | |
|---|---|---|---|---|---|---|
| $d$ | INVERSE | FORWARD | INVERSE | FORWARD | INVERSE | FORWARD |
| 3 | 0.247 | 0.053 | 0.011 | 0.002 | 0.054 | 0.083 |
| 4 | 0.352 | 0.097 | 0.071 | 0.002 | 0.172 | 0.124 |
| 5 | 0.464 | 0.230 | 0.108 | 0.004 | 0.247 | 0.178 |
| 6 | 0.499 | 0.401 | 0.138 | 0.122 | 0.338 | 0.249 |
| 7 | 0.551 | 0.586 | 0.247 | 0.391 | 0.324 | 0.451 |
| 8 | 0.587 | 1.392 | 0.281 | 1.136 | 0.379 | 0.722 |

Table 1: The relative error of the inverse and forward algorithms' estimators for various action sets and dimensions.

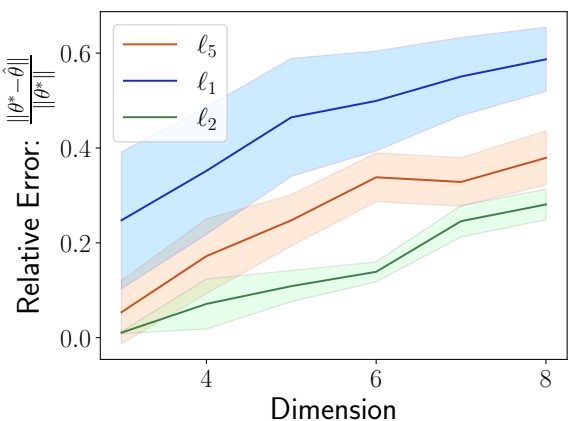

Figure 3: Inverse estimation error as a function of dimension $d$ on each action set. Shaded region represents the standard deviation.

| | MOVIELENS | | AMAZON REVIEWS | |
|---|---|---|---|---|
| $d$ | INVERSE | FORWARD | INVERSE | FORWARD |
| 2 | 0.2859 | 0.0037 | 0.1250 | 0.0018 |
| 4 | 0.3666 | 0.0356 | 0.3646 | 0.0081 |
| 6 | 0.3641 | 0.1401 | 0.4291 | 0.3660 |
| 8 | 0.5030 | 0.4632 | 0.3955 | 0.5949 |

Table 2: Relative error of the inverse estimator on Movie-Lens 25M and the digital music reviews from Amazon Reviews.

bandit setting, is that we limit our demonstrator to being the canonical Phased Elimination algorithm. Moreover, we place assumptions on the density and geometry of the action set for our analysis—weakening these assumptions pose important future directions.

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

| Symbol | Meaning |
|---|---|
| $d$ | Dimension of environment |
| $T$ | Time horizon |
| $L$ | Number of phases |
| $\theta^*$ | True Reward Function Parameter |
| $\theta$ | Demonstrator's Reward Function Parameter |
| $\hat{\theta}$ | Inverse Estimator's Estimated Reward Parameter |
| $\gamma$ | Closeness parameter of action set |
| $a_t$ | Action taken by demonstrator at time $t$ |
| $x_t$ | Reward seen by demonstrator at time $t$ |
| $\eta_t$ | Noise in reward function seen at time $t$ |
| $\mu^*$ | Reward of optimal arm |
| $a^*$ | Optimal action with the highest reward |
| $\chi_1 = \min_{a \in \mathcal{A}} \|a\|_2$ | Smallest-norm action in action set |
| $\chi_2 = \max_{a \in \mathcal{A}} \|a\|_2$ | Largest-norm action in action set |
| $\mathcal{A}_\ell$ | Set of remaining arms at phase $\ell$ |
| $\mathcal{A}_\ell \setminus \mathcal{A}_{\ell-1}$ | Set of eliminated arms before phase $\ell$ |
| $\epsilon_\ell$ | $2^{-\ell}$ used as criteria for elimination |
| $\nu_\ell$ | Error parameter for G-Optimal Design |
| $\delta$ | Probability Parameter for G-Optimal Design |

Table 3: Table of notation used in main paper and proofs

We now collect the proofs of our main results, for which Table 3 summarizes relevant mathematical notation.

## A PHASED ELIMINATION PROOFS

First, we collect properties of the forward algorithm (the phased elimination algorithm) that will be useful for analyzing our inverse estimator. The following lemma, essentially Lemma 6.1 in Esfandiari et al. [2019], shows that the error in the forward algorithm's estimate of the mean reward of any (active) arm decreases as more epochs are executed.

**Lemma A.1** (**Demonstrator's Estimation Error** [Esfandiari et al., 2019]). *Suppose that Algorithm 1 is run, and denote by $\theta_\ell$ the forward algorithm's estimate of the reward parameter $\theta^*$. Denote the "good event"*

$$\mathcal{E}_{\text{good}} := \{|\langle a, \theta_\ell - \theta^* \rangle| \le \nu_\ell \text{ for all } a \in \mathcal{A}_\ell, \ell \in [L]\}. \tag{4}$$

*Then, the good event $\mathcal{E}_{\text{good}}$ occurs with probability at least $1 - |\mathcal{A}|L\delta$.*

*Proof.* Fix an epoch $\ell \in [L]$, an active arm $a \in \mathcal{A}_\ell$, and recall that the failure probability parameter $\delta$ was given as input to Algorithm 1. Then, Lemma 6.1 of Esfandiari et al. [2019] tells us that

$$|\langle a, \theta_\ell - \theta^* \rangle| \le \nu_\ell$$

with probability at least $1 - \delta$. Taking a union bound over all active arms $a \in \mathcal{A}_\ell$ and all $\ell \in [L]$, and noting that $|\mathcal{A}_\ell| \le |\mathcal{A}|$ completes the proof of the lemma. $\square$

Henceforth, we work on the good event $\mathcal{E}_{\text{good}}$ and state and prove a series of simple lemmas. The first such lemma shows that the optimal arm $a^*$ is not eliminated in any of the phases $\ell \in [L]$.

**Corollary A.1.** *The optimal arm $a^*$ remains active throughout, i.e. $a^* \in \mathcal{A}_\ell$ for all $\ell \in [L]$, under the good event $\mathcal{E}_{\text{good}}$.*

*Proof.* From Lemma A.1, for any suboptimal arm $a$, we have

$$\langle a, \theta_\ell \rangle - \langle a^*, \theta_\ell \rangle \le (\langle a, \theta^* \rangle + \nu_\ell) - (\langle a^*, \theta^* \rangle - \nu_\ell) \le 2\iota\epsilon_\ell \le 2\epsilon_\ell.$$

on the good event $\mathcal{E}_{\text{good}}$. Thus, the elimination criterion is *not* satisfied by arm $a^*$ in any epoch $\ell \in [L]$. This completes the proof of the lemma. $\square$

The next lemma shows a related property, i.e. that sufficiently suboptimal arms will be eliminated (and that more suboptimal arms will be eliminated in earlier epochs).

**Lemma 4.1.** *Any arm $a$ close to the optimal arm satisfying*

$$2(1 - \iota)\epsilon_\ell < \langle a^* - a, \theta^* \rangle \leq 4(1 - \iota)\epsilon_\ell \tag{3}$$

*will be in $\mathcal{A}_\ell \setminus \mathcal{A}_{\ell-1}$ with probability at least $1 - |\mathcal{A}|L\delta$. Therefore, with probability at least $1 - |\mathcal{A}|L\delta$, the mean reward of any arm $a \in \mathcal{A}_\ell \setminus \mathcal{A}_{\ell-1}$ is bounded as*

$$\mu^* - 4(1 - \iota)\epsilon_\ell \leq \langle a, \theta^* \rangle \leq \mu^*.$$

*Proof.* Let $b_{\ell-1}$ be the arm that maximizes the reward in epoch $\ell - 1$, i.e. $b_{\ell-1} = \arg\max_{b \in \mathcal{A}_{\ell-1}} \langle b, \theta_{\ell-1} \rangle$. From Lemma A.1, we have any arm $a$ satisfying Equation (3) satisfies

$$
\begin{aligned}
\langle b_{\ell-1} - a, \theta_{\ell-1} \rangle &\leq \langle b_{\ell-1} - a, \theta^* \rangle + 2\nu_{\ell-1} \tag{5} \\
&\leq \langle a^* - a, \theta^* \rangle + 2\nu_{\ell-1} \\
&\leq 4(1 - \iota)\epsilon_\ell + 2\iota\epsilon_{\ell-1} \\
&\leq 2(1 - \iota)\epsilon_{\ell-1} + 2\iota\epsilon_{\ell-1} \tag{6} \\
&= 2\epsilon_{\ell-1}
\end{aligned}
$$

where Equation (5) follows from the good event in Lemma A.1 and Equation (6) follows because $2\epsilon_l = \epsilon_{l-1}$. This implies that arm $a$ will not be eliminated in phase $\ell - 1$. On the other hand, for epoch $\ell$, we have

$$
\begin{aligned}
\langle b_\ell - a, \theta_\ell \rangle &= \langle b_\ell, \theta_\ell \rangle - \langle a, \theta_\ell \rangle \\
&\geq \langle a^*, \theta_\ell \rangle - \langle a, \theta_\ell \rangle \\
&\geq \langle a^* - a, \theta^* \rangle - 2\nu_\ell \tag{7} \\
&= \langle a^* - a, \theta^* \rangle - 2\iota\epsilon_\ell \\
&\geq 2(1 - \iota)\epsilon_\ell - 2\iota\epsilon_\ell \\
&= 2\epsilon_\ell
\end{aligned}
$$

where Equation (7) also follows from the good event in Lemma A.1. Therefore, arm $a$ will be eliminated in phase $\ell$. This proves the first statement of the lemma. The second statement of the lemma, i.e. $\mu^* - 4(1 - \iota)\epsilon_\ell \leq \langle a, \theta^* \rangle \leq \mu^*$, follows by rearranging the original inequalities and noting that $\mu^* := \langle a^*, \theta^* \rangle$.

$\square$

The following is a useful corollary to Lemma 4.1 for arms that are "close" in Euclidean distance to arms satisfying the condition in Lemma 4.1.

**Corollary A.2.** *Consider an arm $a$ that is $\gamma$-close to some arm $b$ in Euclidean distance, i.e. $\|b - a\|_2 \leq \gamma$, such that arm $b$ satisfies*

$$\mu^* - 4(1 - \iota)\epsilon_\ell + \gamma\|\theta^*\|_2^2 \leq \langle a^* - b, \theta^* \rangle \leq \mu^* - 2(1 - \iota)\epsilon_\ell - \gamma\|\theta^*\|_2^2. \tag{8}$$

*Then, under the good event $\mathcal{E}_{\text{good}}$, arm $a$ will be eliminated before phase $L$, i.e. $a \in \mathcal{A}_L \setminus \mathcal{A}_{L-1}$.*

*Proof.* We have that $|\langle b - a, \theta^* \rangle| \leq \gamma\|\theta^*\|_2$ since $\|a - b\|_2 \leq \gamma$. Therefore,

$$
\begin{aligned}
\langle a^* - a, \theta^* \rangle &\leq \langle a^* - b, \theta^* \rangle + \gamma\|\theta^*\|_2 \\
&\leq \mu^* - 4(1 - \iota)\epsilon_\ell
\end{aligned}
$$

Moreover, we have

$$
\begin{aligned}
\langle a^* - a, \theta^* \rangle &\geq \langle a^* - b, \theta^* \rangle - \gamma\|\theta^*\|_2 \\
&\geq \mu^* - 2(1 - \iota)\epsilon_\ell
\end{aligned}
$$

Thus, directly applying Lemma 4.1 shows that arm $a$ will be eliminated.

$\square$

This completes our set of lemmas that work on the good event $\mathcal{E}_{\text{good}}$. Finally, we provide a lemma that characterizes the total number of phases $L$, which is technically a random variable, in terms of a deterministic upper bound that is logarithmic in $T$.

**Lemma A.2.** *The total number of rounds of Algorithm 1 and the total number of phases $L$ exhibit the relationship*

$$\log(T) \leq \log(2\iota^{-2}dJ) + 2\log(2^L) + \log(2).$$

*Here, $J$ is notational shorthand, defined as $J := \left( \frac{|\mathcal{A}|L(L+1)}{\delta} \right)$.*

*Proof.* Let $N_\ell$ be the number of arms played in phase $\ell$. From Lattimore and Szepesvári [2020], we have

$$N_\ell - \frac{d(d+1)}{2} \leq \frac{2d}{\nu_\ell^2} \log\left( \frac{|\mathcal{A}|l(l+1)}{\delta} \right) \tag{9}$$

$$\leq 2\iota^{-2}d \cdot 2^{2l} \left( \frac{|\mathcal{A}|l(l+1)}{\delta} \right)$$

Recall the notational shorthand $J := \left( \frac{|\mathcal{A}|L(L+1)}{\delta} \right)$. We have

$$\log\left( \sum_\ell^{L-1} N_\ell \right) \leq \log\left( \sum_\ell^{L-1} 2\iota^{-2}d \cdot 2^{2l} \cdot (J) + \frac{d(d+1)}{2} \right)$$

$$= \log\left( 2\iota^{-2}d\,(J) \sum_\ell^{L-1} 2^{2l} + \sum_\ell^{L-1} \frac{d(d+1)}{2} \right)$$

$$= \log\left( 2\iota^{-2}d\,(J) \sum_\ell^{L-1} 2^{2l} \right) + \log\left( \frac{\sum_\ell^{L-1} \frac{d(d+1)}{2}}{2\iota^{-2}d\,(J) \sum_\ell^{L-1} 2^{2l}} \right)$$

$$\leq \log\left( 2\iota^{-2}d\,(J) \sum_\ell^{L-1} 2^{2l} \right) + \log(2)$$

$$\leq \log(2\iota^{-2}dJ) + 2\log(2^L) + \log(2)$$

This completes the proof of the lemma. $\qquad\square$

# B INVERSE ESTIMATOR PROPERTIES

The proof of Theorem 4.1 relies on several intermediate lemmas. We first state and prove these lemmas, and then provid the proof of Theorem 4.1.

## B.1 LEMMAS FOR INVERSE ESTIMATOR ANALYSIS

First, we state and prove a simple lemma that upper bounds our normalized inverse estimation error as a function of the condition number of the matrix whose rows constitute the set of selected arms $\mathcal{A}^e$ and the normalized estimation error of the rewards of the arms in $\mathcal{A}^e$.

**Lemma B.1.** *Suppose $r$ and $\hat{r}$ are vectors of the true rewards and estimated rewards for $\mathcal{A}^e$. If the arms in $\mathcal{A}^e$ are linearly independent, the solution to $\hat{\theta} = \arg\min \sum_{a^i \in \mathcal{A}^e} (\hat{r}_i - \langle \theta, a^i \rangle)^2)$ where $\hat{r}_i$ is the estimate reward of $a^i$ satisfies the bound the error in estimation of $\theta$ via*

$$\frac{\left\| \hat{\theta} - \theta^* \right\|_2}{\|\theta\|_2} \leq \text{cond}(\mathcal{A}^e) \frac{\|\hat{r} - r\|_2}{\|r\|_2}.$$

*Proof.* We consider the design matrix $\mathbf{A}$ whose $d$ rows are given by the arms in $\mathcal{A}^e$. More formally, we define

$$
\mathbf{A} = \begin{bmatrix} a^1 \\ a^2 \\ \vdots \\ a^d \end{bmatrix}
$$

where $a^1, \ldots, a^d \in \mathcal{A}^e$. Therefore, the solution to the least squares problem is given by $\hat{\theta} = \arg\min \sum_{a^i \in \mathcal{A}^e} (\hat{r}_i - \langle \theta, a^i \rangle)^2)$ is solved by

$$
\hat{\theta} = (\mathbf{A}^T \mathbf{A})^{-1} \mathbf{A}^T \hat{r} = \mathbf{A}^{-1} \hat{r}, \tag{10}
$$

where the last equality follows because $\mathbf{A}$ is a square matrix and the arms are linearly independent (Lemma 4.2). Therefore, we have

$$
\|\hat{\theta} - \theta^*\|_2 = \|\mathbf{A}^{-1}(r - \hat{r})\|_2
$$
$$
\leq \|\mathbf{A}^{-1}\|_2 \|r - \hat{r}\|_2
$$

Moreover, we have

$$
\|r\|_2 = \|\mathbf{A}\theta^*\|_2 \leq \|\mathbf{A}\|_2 \|\theta^*\|_2.
$$

Combining the inequalities above completes the proof of the lemma. $\qquad\square$

Next, we restate and prove our main technical lemma, which characterizes the condition number of the design matrix $\mathbf{A}$ whose rows consist of the arms in $\mathcal{A}^e$.

**Lemma 4.2 (Condition Number of $\mathcal{A}^e$).** *Let $\chi_2$ and $\chi_1$ be defined as $\chi_2 = \max\limits_{a \in \mathcal{A}} \|a\|_2, \chi_1 = \min\limits_{a \in \mathcal{A}} \|a\|_2$. Suppose that Assumption 4.1 holds, and we can select the action subset $\mathcal{A}^e$ according to Steps 4-6 of Algorithm 2. Then, with probability at least $1 - |\mathcal{A}|L\delta$, the condition number of the matrix whose rows are elements of $\mathcal{A}^e$ satisfies*

$$
\text{cond}(\mathcal{A}^e) \leq \frac{\chi_2 + \gamma\sqrt{d}}{\chi_1 \left[ (2d)^{-\frac{1}{2}} \beta \right] - \gamma\sqrt{d}}.
$$

*Proof.* We will break down the proof of the bound of the condition number into two parts. First, we decompose $\mathbf{A}$ into the following convenient form:

$$
\mathbf{A} = \mathbf{D}\tilde{\mathbf{A}} + \mathbf{N}.
$$

Above, $\mathbf{D}$ is a diagonal matrix such that $D_{i,i} = \|a^i\|_2$, and $\tilde{\mathbf{A}}$ is a matrix such that $i$th row of $\mathbf{A}$, which we denote as shorthand by $v_i$, is $v_i = \frac{\text{proj}(a^i, i)}{\|\text{proj}(a^i, i)\|_2}$. (Recall that $\text{proj}(a^i, i)$ was defined in Section 4.1 and is the projection of the arm $a^i$ onto the plane spanned by the optimal arm $a^*$ and the $i$-th element of the regular simplex $s_i$.) Finally, $\mathbf{N}$ constitutes an "error" matrix term whose $i$-th row is equal to $a^i - \text{proj}(a^i, i)$. We expect $\mathbf{N}$ to be "small" in the sense of operator-norm under Assumption 4.1; we will show this formally shortly.

Since $\text{cond}(\mathcal{A}^e) = \frac{\sigma_{\max}(\mathbf{A})}{\sigma_{\min}(\mathbf{A})}$, it suffices to lower bound $\sigma_{\min}(\mathbf{A})$ and upper bound $\sigma_{\max}(\mathbf{A})$ in order to upper bound the condition number. First, we provide a lower bound on the minimum singular value $\sigma_{\min}(\mathbf{A}_\ell)$. By Weyl's theorem, we have

$$
\sigma_{\min}(\mathbf{A}) = \sigma_{\min}(\mathbf{D}\tilde{\mathbf{A}} + \mathbf{N})
$$
$$
\geq \sigma_{\min}(\mathbf{D}\tilde{\mathbf{A}}) - \sigma_{\max}(\mathbf{N}) \tag{11}
$$

Then, we can upper bound $\sigma_{\max}(\mathbf{N})$ as below:

$$
\sigma_{\max}(\mathbf{N}) = \sqrt{\|\mathbf{N}^\top \mathbf{N}\|_2} \tag{12}
$$
$$
= \sqrt{\max_{x \text{ s.t. } \|x\|_2 = 1} x^\top \mathbf{N}^\top \mathbf{N} x}
$$
$$
\leq \sqrt{d\gamma^2}
$$
$$
= \gamma\sqrt{d}
$$

Above, Equation (12) comes from noticing that the rows of $\mathbf{N}$ have $\ell_2$ norm at most $\gamma$ — this is because $\|a^i - \operatorname{proj}(a^i, i)\|_2 =:$ $\operatorname{dist}(a^i, i) \leq \gamma$, where the last inequality uses part 2 of Assumption 4.1. Thus, we have $\sigma_{\min}(\mathbf{A}) \geq \sigma_{\min}(\mathbf{D}\tilde{\mathbf{A}}) - \gamma\sqrt{d}$. A symmetric argument for the maximum singular value gives us $\sigma_{\max}(\mathbf{A}) \leq \sigma_{\max}(\mathbf{D}\tilde{\mathbf{A}}) + \gamma\sqrt{d}$.

Next, we characterize the minimum and maximum singular values of the product matrix $\mathbf{D}\tilde{\mathbf{A}}$. Starting with the minimum singular value, note that $\sigma_{\min}(\mathbf{D}\tilde{\mathbf{A}}) \geq \sigma_{\min}(\mathbf{D})\sigma_{\min}(\tilde{\mathbf{A}})$. Since $\mathbf{D}$ is a diagonal matrix, we have $\sigma_{\min}(\mathbf{D}) = \min_{i \in [d]} D_{i,i} \geq \min_{a \in \mathcal{A}} \|a\|_2 =: \chi_1$. Therefore, we have

$$\sigma_{\min}(\mathbf{A}) \geq \chi_1 \sigma_{\min}(\tilde{\mathbf{A}}) - \gamma\sqrt{d}.$$

Similarly, for the maximum singular value we have $\sigma_{\max}(\mathbf{D}) = \max_{i \in [d]} D_{i,i} \leq \max_{a \in \mathcal{A}} \|a\|_2 =: \chi_2$. This gives us

$$\sigma_{\max}(\mathbf{A}) \leq \chi_2 \sigma_{\max}(\tilde{\mathbf{A}}) + \gamma\sqrt{d}$$

.

We now only need to analyze the minimum and maximum singular values of $\tilde{\mathbf{A}}$; this forms the technical crux of our proof. Recall that the rows of $\tilde{\mathbf{A}}$ are equal to $v_i := \frac{\operatorname{proj}(a^i, i)}{\|\operatorname{proj}(a^i, i)\|_2}$. Further, define the normalized matrix $\mathbf{B} = \frac{1}{\sqrt{d}}\tilde{\mathbf{A}}$ for convenience. We will characterize the eigenvalues of the matrix $\mathbf{B}^\top \mathbf{B}$, noting that $\sigma_j(\tilde{\mathbf{A}}) = \sqrt{d \cdot \lambda_j(\mathbf{B}^\top \mathbf{B})}$. Note that $(\mathbf{B}^\top \mathbf{B})_{i,j} = \frac{1}{d}\langle v_i, v_j \rangle$, and so

$$(\mathbf{B}^\top \mathbf{B})_{i,i} = \frac{1}{d}$$

for all $i \in [d]$. We now characterize the off-diagonal terms. Note that $\langle v_i, v_j \rangle = 1 - \frac{\|v_i - v_j\|_2^2}{2}$, so it suffices to characterize the terms $\|v_i - v_j\|_2^2$.

We wish to first find the angle between our $\alpha$ vectors. We remind the reader that our $\alpha$ vectors form a $d-1$-dimensional simplex centered at the unit vector $u = \frac{a^*}{\|a^*\|_2}$. We will first find the radius of this simplex, i.e., $\|u - v_i\|_2$. The vectors $u$, $v_i$, and the origin form an isosceles triangle where $u$ and $v_i$ are unit-norm by definition. Therefore, by the Law of Sines

$$\|u - v_i\|_2 = \frac{\sin(\tau(a^i, i))}{\sin\left(\frac{\pi - \tau(a^i, i)}{2}\right)}$$
$$= 2\sin\left(\frac{\tau(a^i, i)}{2}\right)$$

Therefore, we have that the radius of the simplex is $2\sin\left(\frac{\tau(a^i, i)}{2}\right)$, which we will call $\rho$ for now. From Krasnodębski [1971], the angles formed between $u - v_i$ and $u - v_j$ is $\arccos\left(-\frac{1}{d-1}\right)$. Therefore, we have the distance between $v_j$ and $v_i$ satisfies

$$\|v_j - v_i\|_2^2 = \|u - v_i\|_2^2 + \|u - v_j\|_2^2 - 2\|u - v_i\|_2\|u - v_j\|_2 \cos\left(\arccos\left(-\frac{1}{d-1}\right)\right)$$
$$= 2\rho^2 \left(1 - \cos\left(\arccos\left(-\frac{1}{d-1}\right)\right)\right)$$
$$= 2\rho^2 \frac{d}{d-1}$$

Therefore, we have

$$\langle v_i, v_j \rangle = 1 - \frac{\rho^2 d}{d-1} =: \cos(\beta).$$

We have shown that we can decompose the matrix $\mathbf{B}^\top \mathbf{B}$ as $\mathbf{B}^\top \mathbf{B} = \frac{1 - \cos(\beta)}{d} \cdot \mathbf{I} + \frac{\cos(\beta)}{d} \cdot \mathbf{1}\mathbf{1}^\top$. This matrix has maximum eigenvalue equal to $\frac{1 - \cos(\beta)}{d} + \cos(\beta)$ and minimum eigenvalue equal to $\frac{1 - \cos(\beta)}{d}$. Thus, we can upper bound the maximum eigenvalue as

$$\lambda_{\max}(\mathbf{B}^\top \mathbf{B}) = \frac{d-1}{d}\cos(\beta) + \frac{1}{d}$$
$$\leq \frac{d-1}{d} + \frac{1}{d} = 1.$$

Next, we can write the minimum eigenvalue as

$$\lambda_{\min}(\mathbf{B}^\top\mathbf{B}) = \frac{1}{d} - \frac{1}{d}\cos(\beta) \geq \frac{\rho^2}{d-1}. \tag{13}$$

Further, we can lower bound $\rho^2$ on the interval $\tau(a^i, i) \in [-\frac{\pi}{2}, \frac{\pi}{2}]$ via its Taylor expansion as

$$\rho^2 \geq \frac{\tau(a^i, i)^2}{2}. \tag{14}$$

Combining Equation (13) with Equation (14) gives us the following lower bound on the minimum eigenvalue:

$$\lambda_{\min}(\mathbf{B}^\top\mathbf{B}) \geq \frac{\tau(a^i, i)^2}{2d} \geq \frac{\beta^2}{2d}. \tag{15}$$

Thus, we have characterized the minimum and maximum eigenvalues of $\mathbf{B}^\top\mathbf{B}$. Putting all of the steps together, we have

$$\begin{aligned}
\text{cond}(\mathbf{A}) = \frac{\sigma_{\max}(\mathbf{A})}{\sigma_{\min}(\mathbf{A})} &\leq \frac{\chi_1\sigma_{\max}(\tilde{\mathbf{A}}) + \gamma\sqrt{d}}{\chi_2\sigma_{\min}(\tilde{\mathbf{A}}) - \gamma\sqrt{d}} \\
&= \frac{\chi_1\sigma_{\max}(\mathbf{B}) + \gamma d}{\chi_2\sigma_{\min}(\mathbf{B}) - \gamma d} \\
&= \frac{\chi_1\sqrt{\lambda_{\max}(\mathbf{B}^\top\mathbf{B})} + \gamma d}{\chi_2\sqrt{\lambda_{\min}(\mathbf{B}^\top\mathbf{B})} - \gamma d} \\
&\leq \frac{\chi_1 + \gamma d}{\chi_2\left[(2d)^{-\frac{1}{2}}\beta\right] - \gamma d}.
\end{aligned}$$

This completes the proof of the lemma. $\qquad\square$

Next, we restate and prove a lemma that bounds the normalized estimation error of the rewards of the selected arms in $\mathcal{A}^e$.

**Lemma 4.3.** *Let $r$ denote the vector of true rewards $\left\{R_{\theta^*}(a^i)\right\}_{i=1}^d$ and $\hat{r}$ denote a vector of our estimated rewards given by $\{\mu^* - 2(1+\iota)\epsilon_\ell\}_{i=1}^d$. Then, we have $\frac{\|r-\hat{r}\|_2}{\|r\|_2} \leq \frac{4\epsilon_L}{\mu^* - 8\epsilon_L} = \mathcal{O}\left(2^{-L}\right)$ with probability at least $1 - |\mathcal{A}|L\delta$.*

*Proof.* Consider an arm $a^i \in \mathcal{A}^e$ (where $i \in [d]$), and denote $r_i := R_{\theta^*}(a^i)$ as shorthand. Via Lemma 4.1, we have

$$\mu^* - 4(1+\iota)\epsilon_L \leq r_i \leq \mu^*. \tag{16}$$

Denote the corresponding estimator of the mean reward of this arm by $\hat{r}_i := \mu^* - 2(1+\iota)\epsilon_L$. Clearly, we have $|r_i - \hat{r}_i| \leq 2(1+\iota)\epsilon_L$. Thus, we have

$$\|\hat{r} - r\|_2 \leq 2(1+\iota)\epsilon_L\sqrt{d}.$$

Next, we lower bound the denominator $\|r\|_2$. Equation (16) tells us that $|r_i| \geq \mu^* - 4(1+\iota)\epsilon_L$ for every $i \in [d]$. This gives us the lower bound $\|r\|_2 \geq \sqrt{d}(\mu^* - 4(1+\iota)\epsilon_L)$. Putting the pieces together yields

$$\frac{\|r - \hat{r}\|_2}{\|r\|_2} \leq \frac{2(1+\iota)\epsilon_L}{\mu^* - 4(1+\iota)\epsilon_L}.$$

Since $\iota \leq 1$ from Assumption 4.1, we have

$$\frac{2(1+\iota)\epsilon_L}{\mu^* - 4(1+\iota)\epsilon_L} \leq \frac{4\epsilon_L}{\mu^* - 8\epsilon_L} = \mathcal{O}\left(2^{-L}\right).$$

This completes the proof of the lemma. $\qquad\square$

## B.2 PROOF OF THEOREM 4.1

We are now ready to prove Theorem 4.1. First, Lemma B.1 tells us that

$$\frac{\left\|\hat{\theta} - \theta^*\right\|_2}{\|\theta^*\|_2} \le \mathrm{cond}(\mathcal{A}^e) \frac{\|\hat{r} - r\|_2}{\|r\|_2}.$$

From Lemma 4.2, we get

$$\mathrm{cond}(\mathcal{A}^e) \le \frac{\chi_1 + \gamma d}{\chi_2 \left[(2d)^{-\frac{1}{2}}\beta\right] - \gamma d}.$$

Moreover, from Lemma 4.3, we have

$$\frac{\|r - \hat{r}\|_2}{\|r\|_2} \le \frac{4\epsilon_L}{\mu^* - 8\epsilon_L}.$$

Plugging in Assumption 4.1 (which stipulates that $\beta = (3(1 - \iota)\epsilon_L)^{\frac{1}{\omega}}$) and the above bounds gives us

$$\frac{\left\|\hat{\theta} - \theta^*\right\|_2}{\|\theta\|_2} \le \frac{\chi_1 + \gamma d}{\chi_2 \left[(2d)^{-\frac{1}{2}} [3(1 - \iota)\epsilon_L]^{\frac{1}{\omega}}\right] - \gamma d} \cdot \frac{4\epsilon_L}{\mu^* - 8\epsilon_L}$$

$$\le \frac{\chi_1 + \gamma d}{2^{\frac{L(\omega-1)}{\omega}} \cdot \chi_2 \left[(2d)^{-\frac{1}{2}} [3(1 - \iota)]^{\frac{1}{\omega}}\right] - 2^L \gamma d} \cdot \frac{4}{\mu^* - 8\epsilon_L}.$$

It remains to express the above upper bound in terms of the deterministic quantity $T$ of interest (rather than the total number of phases $L$, which is random). For this, Lemma A.2 tells us that

$$\log(T) \le \log(2\iota^{-2}dJ) + 2\log(2^L) + \log(2).$$

Using this, we have

$$\left[\frac{T}{4\iota^{-2}dJ}\right]^{\frac{1}{2}} \le 2^L$$

$$\implies 2^{\frac{L(1-\omega)}{\omega}} \le \left[\frac{T}{4\iota^{-2}dJ}\right]^{\frac{1-\omega}{2\omega}}.$$

Plugging this into the upper bound then gives us

$$\frac{\left\|\hat{\theta} - \theta^*\right\|_2}{\|\theta\|_2} \le \frac{\chi_1 + \gamma d}{\left[\frac{T}{4\iota^{-2}dJ}\right]^{\frac{\omega-1}{2\omega}} \chi_2 \left[(2d)^{-\frac{1}{2}} [3]^{\frac{1}{\omega}}\right] - 2^L \gamma d} \cdot \frac{4}{\mu^* - 8\epsilon_L}.$$

(17)

Given $\gamma \le \frac{\epsilon_{\bar{L}}}{\|\theta^*\|_2 d} \le \frac{2^{-L}}{\|\theta^*\|_2 d}$ from Assumption 4.1 and $\frac{4}{\mu^* - 8\epsilon_L} = \mathcal{O}(1)$, and noting that the term $\left[\frac{T}{4\iota^{-2}dJ}\right]^{\frac{1-\omega}{2\omega}}$ is increasing in $T$, we get

$$\frac{\left\|\hat{\theta} - \theta^*\right\|_2}{\|\theta\|_2} = \mathcal{O}\left(\frac{\chi_1 d^{\frac{2\omega-1}{2\omega}} J^{\frac{\omega-1}{2\omega}}}{\chi_2 T^{\frac{\omega-1}{\omega}}}\right).$$

This completes the proof of the theorem. □

## C PROOF OF LOWER BOUND (THEOREM 5.1)

In this section, we provide the proof of Theorem 5.1, which leverages the classical Le-Cam binary testing approach [LeCam, 1973] between a null instance and a random alternative instance. We will actually show the lower bound on estimation error assuming access to *both* the sequence of actions and observed rewards by the forward algorithm, as additionally observing rewards only makes the estimation problem easier.

Formally, we establish two bandit instances:

1. The first instance $\mathcal{M}$ is one in which the linear reward parameter is the true parameter of interest $\theta^*$.

2. The second random instance $\mathcal{M}'(v)$ is one in which the linear reward parameter is given by $\theta'(v) = \theta^* - \epsilon v$, where $\epsilon > 0$ is a parameter that will be chosen appropriately at a later point, and $v \sim \text{Unif}(S^{d-1})$, i.e. $v$ is chosen uniformly at random from the $d$-dimensional unit sphere. Eventually, we will take an expectation over the binary testing error.

Before proceeding with the proof, we set up some more relevant notation. Let $\mathcal{E}_T$ denote the observed sequence of action-reward pairs $(a_1, r_1), \ldots, (a_T, r_T)$ (which is random), and $\mathcal{F}_T$ denote the associated sigma-algebra of possible events. Further, for any arm $a$ let $\mathcal{V}(a)$ and $\mathcal{V}'_v(a)$ denote the associated reward distributions under bandit instances $\mathcal{M}$ and $\mathcal{M}'(v)$ respectively. For convenience, we will assume that the noise in the rewards is drawn from an isotropic Gaussian distribution, meaning that $\mathcal{V}(a) = \mathcal{N}(\langle \theta^*, a \rangle, \mathbf{I})$ and $\mathcal{V}'_v(a) = \mathcal{N}(\langle \theta'(v), a \rangle, \mathbf{I})$. Finally, we denote $\mathbb{E}_0[\cdot], \mathbb{E}'_v[\cdot]$ as the expectations over all randomness in the observation $\mathcal{E}_T$ induced by the bandit instance $\mathcal{M}, \mathcal{M}'$ respectively, and $\mathbb{E}[\cdot]$ will denote any additional expectations, typically to be taken over the randomness in the parameter $v$ only. Further, let $\mathbb{P}_0[\cdot], \mathbb{P}'_v[\cdot]$ denote the probability distributions over the observation $\mathcal{E}_T$ under bandit instances $\mathcal{M}, \mathcal{M}'(v)$ respectively. Finally, we use $D_{\text{KL}}(\cdot, \cdot)$ to denote the Kullback-Liebler divergence between two probability distributions.

Note that $\hat{\theta}$ can only be a functional of the observation $\mathcal{E}_T$. Therefore, for any fixed $v \in S^{d-1}$, the LeCam method gives us

$$\max \left\{ \mathbb{E} \left[ \|\hat{\theta} - \theta^*\|_2 \right], \mathbb{E}' \left[ \|\hat{\theta} - \theta'(v)\|_2 \right] \right\} \geq \frac{1}{2} \|\epsilon v\|_2 \left( 1 - \|\mathbb{P}_0 - \mathbb{P}'_v\|_{\text{TV}} \right)$$
$$= \frac{\epsilon}{2} \left( 1 - \|\mathbb{P}_0 - \mathbb{P}'_v\|_{\text{TV}} \right),$$

where the last equality follows because $v \in S^{d-1}$. Taking a further expectation over $v \sim \text{Unif}(S^{d-1})$ and using linearity of expectation yields

$$\mathbb{E} \left[ \max \left\{ \mathbb{E} \left[ \|\hat{\theta} - \theta^*\|_2 \right], \mathbb{E}' \left[ \|\hat{\theta} - \theta'(v)\|_2 \right] \right\} \right] \geq \mathbb{E} \left[ \frac{\epsilon}{2} \left( 1 - \|\mathbb{P}_0 - \mathbb{P}'_v\|_{\text{TV}} \right) \right]$$
$$= \frac{\epsilon}{2} \left( 1 - \mathbb{E} \left[ \|\mathbb{P}_0 - \mathbb{P}'_v\|_{\text{TV}} \right] \right). \tag{18}$$

Therefore, it suffices to upper bound the term $\mathbb{E} \left[ \|\mathbb{P}_0 - \mathbb{P}'_v\|_{\text{TV}} \right]$. First, we consider the total variation distance $\|\mathbb{P}_0 - \mathbb{P}'_v\|_{\text{TV}}$ for a fixed $v$. By the definition of total variation distance, we have $\|\mathbb{P}_0 - \mathbb{P}'_v\|_{\text{TV}} := \sup_{\mathcal{E}_T \in \mathcal{F}_T} |\mathbb{P}_0(\mathcal{E}_T) - \mathbb{P}'_v(\mathcal{E}_T)|$. Then, an adaptation of Lemma 19 of Kaufmann et al. [2014] gives us

$$\sup_{\mathcal{E}_T \in \mathcal{F}_T} |\mathbb{P}_0(\mathcal{E}_T) - \mathbb{P}'_v(\mathcal{E}_T)| \leq \sum_{t=1}^{T} \mathbb{E}_0 \left[ D_{\text{KL}}(\mathcal{V}(a_t), \mathcal{V}'_v(a_t)) \right]. \tag{19}$$

Next, note that for any fixed $a$, we have that $\mathcal{V}(a) = \mathcal{N}(\langle \theta^*, a \rangle, \mathbf{I})$ and $\mathcal{V}'_v(a) = \mathcal{N}(\langle \theta'(v), a \rangle, \mathbf{I})$. Therefore, we have $D_{\text{KL}}(\mathcal{V}(a), \mathcal{V}'_v(a)) = \frac{\epsilon^2}{2} (\langle a, v \rangle)^2$. Plugging this into Equation (19) gives us

$$\|\mathbb{P}_0 - \mathbb{P}'_v\|_{\text{TV}} \leq \sum_{t=1}^{T} \mathbb{E}_0 \left[ \frac{\epsilon^2}{2} (\langle a_t, v \rangle)^2 \right]$$
$$= \frac{\epsilon^2}{2} \cdot v^\top \mathbb{E}_0 \left[ \sum_{t=1}^{T} a_t a_t^\top \right] v.$$

Henceforth, we denote $\overline{M}_T := \mathbb{E}_0 \left[ \sum_{t=1}^{T} a_t a_t^\top \right]$ as the expected Gram matrix composed of the actions $a_1, \ldots, a_T$. Note that $\overline{M}_T$ is a deterministic quantity. We leverage the following key fact that was proved by Banerjee et al. [2022], restated below: for some universal positive constant $C > 0$ that does not depend on $T$ or $d$, we have

$$\lambda_{\max}(\overline{M}_T) \leq CT$$
$$\lambda_i(\overline{M}_T) \leq \frac{CT}{d} \text{ for all } i > 1. \tag{20}$$

We will leverage this fact to complete the proof of our main result. Let $\{u_i\}_{i=1}^{d}$ denote the unit-normalized eigenvectors of $\overline{M}_T$, and let $v = \sum_{i=1}^{d} \alpha_i u_i$ (note that while $\{\alpha_i\}_{i=1}^{d}$ are random variables, the eigenvectors $\{u_i\}_{i=1}^{d}$ are deterministic).

Then, taking an expectation on our point-wise bound on the total variation distance over $v \sim \text{Unif}(S^{d-1})$ yields

$$\mathbb{E}\left[\|\mathbb{P}_0 - \mathbb{P}'_v\|_{\text{TV}}\right] \leq \frac{\epsilon^2}{2} \cdot \mathbb{E}\left[v^\top \overline{M}_T v\right]$$
$$= \frac{\epsilon^2}{2d} \cdot \text{trace}(\overline{M}_T),$$

where the last equality follows because $v$ is uniformly distributed on the sphere, and therefore $\mathbb{E}[vv^\top] = \frac{1}{d}\mathbf{I}$. We then plug in Equation (20), which gives us $\text{trace}(\overline{M}_T) := \sum_{i=1}^d \lambda_i(\overline{M}_T) \leq 2CT$. Ultimately, we get

$$\mathbb{E}\left[\|\mathbb{P}_0 - \mathbb{P}'_v\|_{\text{TV}}\right] \leq \frac{\epsilon^2 CT}{2d}. \tag{21}$$

Substituting this in Equation (18) ultimately gives us

$$\mathbb{E}\left[\max\left\{\mathbb{E}\left[\|\hat{\theta} - \theta^*\|_2\right], \mathbb{E}'\left[\|\hat{\theta} - \theta'(v)\|_2\right]\right\}\right] \geq \frac{\epsilon}{2}\left(1 - \frac{\epsilon^2 CT}{2d}\right).$$

Finally, we select $\epsilon = \sqrt{\frac{d}{2C'T}}$ for some sufficiently large constant $C' > C$. This yields the lower bound

$$\mathbb{E}\left[\max\left\{\mathbb{E}\left[\|\hat{\theta} - \theta^*\|_2\right], \mathbb{E}'\left[\|\hat{\theta} - \theta'(v)\|_2\right]\right\}\right] \geq \sqrt{\frac{d}{8C'T}}\left(1 - \frac{C}{C'}\right) = \Omega\left(\sqrt{\frac{d}{T}}\right).$$

This is the desired statement and completes the proof of the theorem. $\qquad\square$

# D    PROOF OF LEMMA 4.4

Recall that Assumption 4.1 is parametrized by scalars $(\omega, \gamma)$ (and through $\omega$, the scalar $\beta$). We construct a family of action set/$\theta^*$ pairs in two dimensions and claim that each pair in the family satisfies Assumption 4.1. We denote the coordinate system using $(x, y) \in \mathbb{R}^2$. Our instance is parametrized by an angle parameter $\kappa$.

**Constructing the action set:**

- Choose $\theta^*$ forming angle $\kappa$ with the vector $(1, 0)$. Set $G = \cos(\kappa)\|\theta^*\|_2 - 3(1 - \iota)\epsilon_L$ for convenience.
- Construct action set $\mathcal{A}$ by including the convex hull[6] of the following points: $(-1, 0)$, $(1, 0)$, $(0, 1)$, $(0, -1)$ as well as the points $\left(\frac{G\cos(\beta)}{\cos(\kappa+\beta)\|\theta^*\|_2}, \frac{G\sin(\beta)}{\cos(\kappa+\beta)\|\theta^*\|_2}\right)$ and $\left(\frac{G\cos(-\beta)}{\cos(\kappa-\beta)\|\theta^*\|_2}, \frac{G\sin(-\beta)}{\cos(\kappa-\beta)\|\theta^*\|_2}\right)$.

See Figure 4 for an illustration of the set. Before we characterize this set, we will define a helper function

**Definition D.1.** *The function* $\text{atan2}(y, x)$ *is defined as*

$$\text{atan2}(y, x) = \begin{cases} \arctan\left(\frac{y}{x}\right) & \textit{if } x > 0 \\ \arctan\left(\frac{y}{x}\right) + \pi & \textit{if } y \geq 0 \textit{ and } x < 0 \\ \arctan\left(\frac{y}{x}\right) - \pi & \textit{if } y < 0 \textit{ and } x < 0 \\ +\frac{\pi}{2} & \textit{if } y > 0 \textit{ and } x = 0 \\ -\frac{\pi}{2} & \textit{if } y < 0 \textit{ and } x = 0 \\ \textit{undefined} & \textit{if } y = 0 \textit{ and } x = 0 \end{cases}$$

The two crucial and readily verifiable properties about this set that will be used in the sequel are that

1. All arms $(x, y) \in \mathcal{A} \setminus \{(1, 0)\}$ satisfy

$$\cos(\kappa + \text{atan2}(y, x))\|\theta^*\|_2\sqrt{x^2 + y^2} < \cos(\kappa)\|\theta^*\|_2.$$

2. $\mathcal{A}$ contains the points $P_1 = \left(\frac{G\cos(\beta)}{\cos(\kappa+\beta)\|\theta^*\|_2}, \frac{G\sin(\beta)}{\cos(k+\beta)\|\theta^*\|_2}\right)$ and $P_3 = \left(\frac{G\cos(-\beta)}{\cos(\kappa-\beta)\|\theta^*\|_2}, \frac{G\sin(-\beta)}{\cos(k-\beta)\|\theta^*\|_2}\right)$.

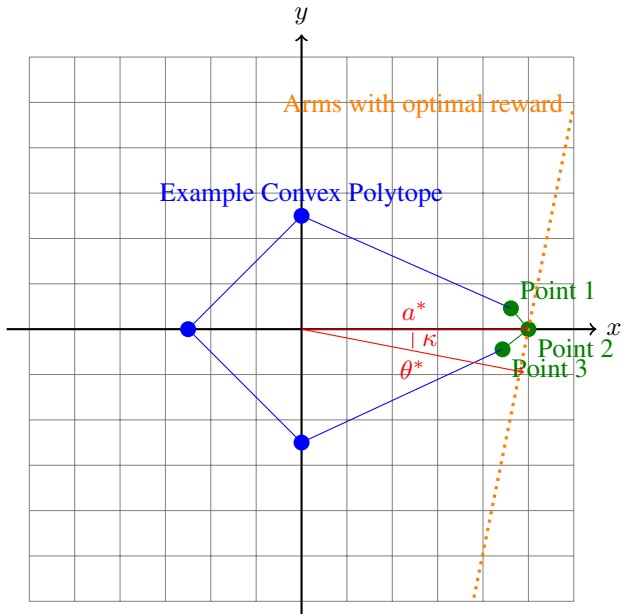

Figure 4: Example Configuration of action set detailed by the proof for Lemma 4.4. The green points are the three points referenced by the proof, the orange line is the line of vectors with the same optimal reward as the optimal Point 2, and the blue lines are example continuations of drawing the convex hull of the action set that satisfy Assumption 3.1. These are done when $\kappa = .2, L = 5,$ and $\beta = .1$.

**Lemma D.1.** *Provided*

$$\kappa \in \left[ \max\left( -\cos^{-1}\left( \frac{3(1-\iota)\epsilon_L}{\|\theta^*\|_2} \right), \cos^{-1}(0) + \beta - \pi \right), \right.$$
$$\left. \min\left( \cos^{-1}\left( \frac{3(1-\iota)\epsilon_L}{\|\theta^*\|_2} \right), \cos^{-1}(0) - \beta \right) \right],$$

*the pair $(\theta^*, \mathcal{A})$ constructed above (or any rotation thereof) satisfies Assumption 4.1.*

*Proof.* We now verify several claims, which when taken together will prove the lemma.

**Claim 1: The optimal arm is $a^* = P_2 = (1, 0)$:** Recall that $\beta = (3(1-\iota)\epsilon_L)^{\frac{1}{\omega}}$, and that every point $(x, y)$ in the action set satisfies $\cos(\kappa + \operatorname{atan2}(y, x))\|\theta^*\|_2 \sqrt{x^2 + y^2} < \cos(\kappa)\|\theta^*\|_2$. Any arm $(x, y) \in \mathcal{A}$ forms an angle of $\kappa + \operatorname{atan2}(y, x)$ with the reward vector $\theta^*$. They also have magnitude of $\sqrt{x^2 + y^2}$. Therefore, their reward is

$$\cos(\kappa + \operatorname{atan2}(y, x))\|\theta^*\|_2 \sqrt{x^2 + y^2}.$$

The reward of the optimal arm by definition is also

$$\cos(\kappa)\|\theta^*\|_2.$$

Therefore, by the first constraint, we have that any arm in the action set has reward less than the optimal arm.

**Claim 2: Point $P_3$ forms an angle of $\beta$ with $a^* = (1, 0)$, thereby satisfying the first item of Assumption 4.1:** We can explicitly character the angle $P_3$ forms with $a^*$ by the following

---

[6]Any discretization of the boundary of this set also suffices.

$$\arccos\left(\frac{\frac{G\cos(-\beta)}{\cos(k-\beta)n}}{\sqrt{\left(\frac{G\sin(-\beta)}{\cos(k-\beta)n}\right)^2+\left(\frac{G\cos(-\beta)}{\cos(k-\beta)n}\right)^2}}\right) = \arccos\left(\frac{\frac{G\cos(-\beta)}{\cos(k-\beta)\|\theta^*\|_2}}{G\sqrt{\left(\frac{\sin(-\beta)}{\cos(k-\beta)\|\theta^*\|_2}\right)^2+\left(\frac{\cos(-\beta)}{\cos(k-\beta)\|\theta^*\|_2}\right)^2}}\right)$$

$$= \arccos\left(\frac{\frac{G\cos(-\beta)}{\cos(k-\beta)\|\theta^*\|_2}}{\frac{G}{\cos(k-\beta)\|\theta^*\|_2}\sqrt{\left(\sin(-\beta)\right)^2+\left(\cos(-\beta)\right)^2}}\right)$$

$$= \arccos\left(\frac{\cos(-\beta)}{\sqrt{\left(\sin(-\beta)\right)^2+\left(\cos(-\beta)\right)^2}}\right)$$

$$= \beta$$

Therefore, $P_3$ forms an angle of $\beta$ with $a^*$. Similar logic holds for proving Point 1 forms an angle of $\beta$ with $a^* = (1,0)$.

**Claim 3: The second item of Assumption 4.1 is satisfied:** We will now also prove that the second constraint from Assumption 4.1 holds in this setting. We will evaluate the reward of Point 1. Point 1 forms an angle of $\beta$ with the optimal arm $a^*$ and, thus, forms an angle of $\beta + \kappa$ with $\theta^*$. Moreover, the $\ell_2$ norm of Point 1 is

$$\left|\frac{(\cos(\kappa)\|\theta^*\|_2 - 3(1-\iota)\epsilon_L)}{\cos(\kappa+\beta)\|\theta^*\|_2}\right|.$$

Given the restriction on $\kappa$, we have that $\frac{(\cos(\kappa)\|\theta^*\|_2 - 3(1-\iota)\epsilon_L)}{\cos(\kappa+\beta)\|\theta^*\|_2}$ is strictly positive. Since

$$-\arccos\left(\frac{3(1-\iota)\epsilon_L}{\|\theta^*\|_2}\right) \le \kappa \le \arccos\left(\frac{3(1-\iota)\epsilon_L}{\|\theta^*\|_2}\right),$$

the numerator is positive. Moreover, since $\arccos(0) - \beta - \pi \le \arccos(0) - \beta$ the denominator is positive. Therefore, its reward is

$$\frac{(\cos(\kappa)\|\theta^*\|_2 - 3(1-\iota)\epsilon_L)}{\cos(\kappa+\beta)\|\theta^*\|_2}\|\theta^*\|_2\cos(\beta+\kappa) = \cos(\kappa)\|\theta^*\|_2 - 3(1-\iota)\epsilon_L$$

$$= \mu^* - 3(1-\iota)\epsilon_L$$

We now do this similarly for Point 3. Point 3 forms an angle of $-\beta$ with the optimal arm $a^*$ and, thus, forms an angle of $\kappa - \beta$ with $\theta^*$. Moreover, the $\ell_2$ norm of Point 1 is

$$\left|\frac{(\cos(\kappa)\|\theta^*\|_2 - 3(1-\iota)\epsilon_L)}{\cos(\kappa-\beta)\|\theta^*\|_2}\right|.$$

Given the restrictions on $\kappa$, the value $\frac{(\cos(\kappa)\|\theta^*\|_2 - 3(1-\iota)\epsilon_L)}{\cos(\kappa-\beta)}$ is strictly positive. Since

$$-\arccos\left(\frac{3(1-\iota)\epsilon_L}{\|\theta^*\|_2}\right) \le \kappa \le \arccos\left(\frac{3(1-\iota)\epsilon_L}{\|\theta^*\|_2}\right),$$

the numerator is positive. Moreover, since $\arccos(0) + \beta - \pi \le \arccos(0) + \beta$, the denominator is positive. Therefore, its reward is Therefore, its reward is

$$\frac{(\cos(\kappa)\|\theta^*\|_2 - 3(1-\iota)\epsilon_L)}{\cos(\kappa-\beta)\|\theta^*\|_2}\|\theta^*\|_2\cos(\kappa-\beta) = \cos(\kappa)\|\theta^*\|_2 - 3(1-\iota)\epsilon_L$$

$$= \mu^* - 3(1-\iota)\epsilon_L$$

**Claim 4: The action set is sufficiently dense as in** $\mathrm{dist}(a^i, i) \le \gamma \le \frac{\epsilon_{\bar{L}}}{\|\theta^*\|_2 d}$.

$$\mathrm{dist}(a^i, i) = \|\mathrm{proj}(a^i, i) - a^i\|_2$$

$$= \|a^i - a^i\|_2$$

$$= 0$$

$$\le \gamma \tag{22}$$

$\square$

# E   IMPLEMENTATION DETAILS FOR PHASED ELIMINATION USED IN EXPERIMENTS

---

**Algorithm 3:** Phased Elimination

---

**Input :** $\delta$ (probability parameters), $L$ (number of phases),
$\{\nu_1, \ldots, \nu_L\}$ (error parameters)

**Result:** $a_1, \ldots, a_T$

1   $\ell \leftarrow 0$
2   $\mathcal{A}_1 \leftarrow \mathcal{A}$
3   $t_\ell \leftarrow 0$
4   **while** $\ell < L$ **do**
5      $\varepsilon_\ell \leftarrow 2^{-\ell}$
6      $\pi_\ell \leftarrow$ G-Optimal design of $\mathcal{A}_\ell$ with $\delta$ and $\nu_\ell$
7      $N_\ell \leftarrow 0$
8      **for** $a \in \mathcal{A}_\ell$ **do**
9          $N_\ell(a) \leftarrow \left\lceil \frac{2d\pi_\ell(a)}{\nu_\ell^2} \log\left(\frac{k\ell(\ell+1)}{\delta}\right) \right\rceil$
10         Play action $a$ for $N_\ell(a)$ rounds
11         $N_\ell \leftarrow N_\ell + N_\ell(a)$
12      **end**
13      $V_\ell \leftarrow \sum_{a \in \mathcal{A}_\ell} \pi_\ell(a) aa^\top$
14      $\theta_\ell \leftarrow V_l^{-1} \sum_{t=t_\ell}^{t_\ell+N_\ell} a_t x_t$
15      $\mathcal{A}_{\ell+1} \leftarrow \{a \in \mathcal{A}_\ell \text{ s.t. } \max_{b \in \mathcal{A}_\ell}(\langle \theta_\ell, b - a \rangle) \leq 2\varepsilon_l\}$
16      $t_\ell \leftarrow t_\ell + T_\ell$
17      $\ell \leftarrow \ell + 1$
18   **end**

---

Algorithm 3 formally describes the implementation of Phased Elimination used in our experiments. The behavior of this implementation only differs from Algorithm 1 in the choice of stopping criteria; here, we stop after a maximum number of phases, while Algorithm 1 fixes $T$ and allows $L$ to vary. Line 6 is computed via a convex program with Gurobi solver [Gurobi Optimization, LLC, 2023].

