# OpenReview forum: "One Shot Inverse Reinforcement Learning for Stochastic Linear Bandits"
_auai.org/UAI/2024/Conference — UAI 2024 poster_

### Official Review · Reviewer_nYqV · 2024-03-21

**Q2-1 Originality-Novelty:** 2
**Q2-2 Correctness-Technical Quality:** 3
**Q2-5 Clarity Of Writing:** 2

**Q1 Summary And Contributions:**

The article proposes a new way to recover the reward in the stochastic linear bandit based on the idea of phased elimination. Its main contributions are:

1. Propose a new method for reward learning in stochastic linear bandit.

2. Provide detailed proof of its method.

3. Conduct some experiments to clarify its method.

**Q2-3 Extent To Which Claims Are Supported By Evidence:**

3: Good: the main claims are supported by convincing evidence (in the form of adequate experimental evaluation, proofs, (pseudo-)code, references, assumptions).

**Q2-4 Reproducibility:**

2: Fair: key resources (e.g. proofs, code, data) are unavailable but key details (e.g. proof sketches, experimental setup) are sufficiently well-described for an expert to confidently reproduce the main results.

**Q3 Main Strengths:**

1. The article expresses its algorithm procedure clearly and provides detailed proof.
2. The article does some experiments to clarify its method.

**Q4 Main Weakness:**

1. Though the article gives an instance-formulation of assumption 4.1, I find no content about whether assumption 4.1 is easy to achieve, which is not obvious to me. Only the existence of an instance-formulation is not enough.
2. The article claims inverse reinforcement learning in its title while it fails to express the IRL problem for stochastic linear bandit clearly. For instance, why the best reward is needed.
3. The result of the experiment is not clearly analyzed.

**Q5 Detailed Comments To The Authors:**

Below are my comments on the paper:

1. I recommend the author define the IRL problem formally and clearly in the problem formulation section.

2. The experimentation section can be introduced more clearly. Maybe compare the performance of policy under the learned reward with some other IRL methods.

**Q9 Complying With Reviewing Instructions:**

Yes

---

> ### Author Rebuttal · Authors · 2024-04-05
>
> We very much appreciate your thoughtful review and positive assessment of our paper. Below, we respond point-to-point to each of our concerns and suggestions. Please let us know if you have any follow-up questions, which we would be happy to answer.
>
> ## Weaknesses
>
> > W1. Though the article gives an instance-formulation of assumption 4.1, I find no content about whether assumption 4.1 is easy to achieve, which is not obvious to me. Only the existence of an instance-formulation is not enough.
>
> A1. Thanks for the constructive feedback. We wanted to clarify a couple of points about Lemma 4.4 – first, while this reads as an “existence” result, the proof is constructive, and we identify concrete action sets in the proof. Second, Lemma 4.4 shows that specific action sets can be constructed for any value of $\omega$, which is stronger than simply satisfying Assumption 4.1. Assumption 4.1 can be satisfied for some value of $\omega$ for commonly encountered action sets such as $\ell_p$ balls.
>
> &nbsp;
>
> > W2. The article claims inverse reinforcement learning in its title while it fails to express the IRL problem for stochastic linear bandit clearly. For instance, why the best reward is needed.
>
> A2. While knowledge of the optimal reward is a well-known identifiability issue in the IRL literature (see Guo et al. ‘21), we have laid out a way to slightly alter our inverse learner such that we relax the assumption on the knowledge of $\mu^*$. We will include this modification in the camera-ready version.
> Thank you for the suggestion to include a formal definition of the IRL problem for stochastic linear bandits. Recall that the **main elements of IRL** are defined below:
> 1. Knowns: state space $\mathcal{S}$, action space $\mathcal{A}$, transition model $P(s’|s,a)$.
> 2. Unknown: Reward function parameters, i.e. $r(s,a)$
> 3. Input: Demonstration: $(s_1, a_1, \ldots s_H, a_H)$ using policy $\pi$ (commonly considered to be the optimal policy $\pi^*$). Often, multiple such demonstrations are considered.
> 4. Objective: to estimate $r(s,a)$ from multiple demonstrations.
>
>
> Similarly, we define the setting of **inverse learning for stochastic linear bandits (SLB)** below:
>
> 1. Knowns: There is no state space; the action space is the action set $\mathcal{A}$, and the functional form of the reward is linear, i.e. $r(a) = \langle a, \theta^*\rangle$.
> 2. Unknown: Reward function parameter $\theta$
> 3. Input: Sequence of actions $(a_1,...a_T)$ taken by a stochastic linear bandit demonstrator
> 4. Objective: to estimate $\theta^*$ from a single demonstration
> We will expand the formality of the IRL definition in Section 3.3 along these lines. We are happy to take additional suggestions to improve the clarity of exposition of the problem formulation.
>
> &nbsp;
>
> > W3. The result of the experiment is not clearly analyzed.
>
> A3. We would be happy to take concrete suggestions to improve the exposition of the experiments section.
>
> &nbsp;
>
> ## References
> Wenshuo Guo, Kumar Krishna Agrawal, Aditya Grover, Vidya Muthukumar, and Ashwin Pananjady. Learning from an exploring demonstrator: Optimal reward estimation for bandits. arXiv preprint arXiv:2106.14866, 2021.

---

### Official Review · Reviewer_69Gt · 2024-03-21

**Q2-1 Originality-Novelty:** 3
**Q2-2 Correctness-Technical Quality:** 4
**Q2-5 Clarity Of Writing:** 3

**Q10 Ethical Concerns:**

No.

**Q1 Summary And Contributions:**

This paper addresses the problem of Inverse Reinforcement Learning for Stochastic Linear Bandits, in which the inverse learner is given access to a single demonstration (one shot), where the target problem is a specific form of bandit (stochastic linear bandit) and the forward bandit algorithm is assumed to be the specific algorithm of Phase Elimination. The paper presents a novel inverse learning algorithm for this problem based on careful theoretical considerations and provides a theoretical guarantee on the accuracy of the inverse learner in the form of an upper bound on the relative error of the estimated reward of the forward learner as well as a nearly matching lower bound.  The guarantees are based on certain assumptions about the set-up of the problem and the property of the target bandit problem and viability of these assumptions is discussed. Experiments are provided, using synthetic and semi-synthetic data (the latter based on MovieLens), which confirm the behavior expected by the theory.

**Q2-3 Extent To Which Claims Are Supported By Evidence:**

3: Good: the main claims are supported by convincing evidence (in the form of adequate experimental evaluation, proofs, (pseudo-)code, references, assumptions).

**Q2-4 Reproducibility:**

4: Excellent: key resources (e.g. proofs, code, data) are available and key details (e.g. proof sketches, experimental setup) are comprehensively described for competent researchers to confidently and easily reproduce the main results.

**Q3 Main Strengths:**

A solid technical paper on an interesting and potentially important problem.

The theoretical results are non-trivial and of high quality, establishing a near optimal guarantee albeit some technical assumptions.

Experiments are well-designed and well-described.

Thorough technical details are provided for the theory and the experiments.

**Q4 Main Weakness:**

The set up of fixing the learner to be a specific algorithm is rather limiting (but I appreciate the challenge involved in broadening that and this is a good start).

Assumption of access to the optimal arm and the best reward seems somewhat impractical (but an account is given that it can be relaxed in practice).

The technical content of the paper is dense and a bit hard for non-experts to follow.

**Q5 Detailed Comments To The Authors:**

Further explanation on why the assumption of access to the optimal arm and the best reward is essential for solving the problem, how realistic is the relaxation described in Footnote 3, and any insights on whether these assumptions can be removed altogether in the future would be good to have.

A definition of "count number"(or a reference) should be given for those readers unfamiliar with this area.

There are some minor typos, etc., which should be corrected in the final version.

**Q9 Complying With Reviewing Instructions:**

Yes

---

> ### Author Rebuttal · Authors · 2024-04-05
>
> We very much appreciate your thoughtful review and positive assessment of our paper. Below, we respond point-to-point to each of our concerns and suggestions. Please let us know if you have any follow-up questions, which we would be happy to answer.
>
> ## Weaknesses
>
> > W1. The set up of fixing the learner to be a specific algorithm is rather limiting (but I appreciate the challenge involved in broadening that and this is a good start).
>
> A1. Thanks for the feedback and your appreciation of the challenge in broadening the assumption of fixing the bandit algorithm. We wish to add some additional context here: the only prior work on inverse learning from a single demonstration (Guo et al., 2021) also required specific bandit algorithm(s) to conduct their analysis, even in the simpler MAB case. Our paper demonstrates that inverse learning is even more challenging for the structured and continuous-armed bandit setting. Nevertheless, we believe weaker objectives than exact inverse estimation (such as ranking of arms) might be possible if we do not assume knowledge of a specific algorithm.
>
> &nbsp;
>
> > W2. Assumption of access to the optimal arm and the best reward seems somewhat impractical (but an account is given that it can be relaxed in practice).
>
> A2. We elaborate here on how relaxing access to $a^*$ and $\mu^*$ would be possible. First, note that $a^*$ can be approximately estimated from the forward demonstrations by either selecting the last arm ($a_T$) or taking a suitable average over all the arms in the last eliminated set $\sum_{a \in \mathcal{A_L} \setminus \mathcal{A_{L-1} } }a$. We expect only an additional estimation error of $a^*$ to arise in the statement of Theorem 4.1, and that this error term would be $\mathcal{O}(d/\sqrt{T})$, and so will not dominate. Second, we have laid out a way to slightly alter our inverse learner such that we relax the assumption on the knowledge of $\mu^*$ by estimating the suboptimality gaps rather than the rewards of the arms $a^1,\ldots,a^d$. This is reminiscent of the strategy that was followed in (Guo et al., 2021), but the role of the suboptimality gaps is different. In particular, our modified procedure can collect estimates of the suboptimality gaps for $d+1$ arms (instead of estimates of the mean rewards of $d$ arms). Then, we can perform a similar linear regression procedure to learn not only the $d$ parameters in $\theta^*$ but also $\mu^* = \langle a^*, \theta^* \rangle$. This turns out not to add extra estimation error beyond a multiplicative constant factor. If it is believed to be of interest to readers, we would be happy to expand on the details of this modified procedure in the camera-ready version.
>
> &nbsp;
>
> > W3. The technical content of the paper is dense and a bit hard for non-experts to follow.
>
> A3. Thanks for the feedback. We would be happy to take into account any concrete suggestions to improve the readability of the paper for non-experts.
>
> &nbsp;
>
> ## Questions
>
> > Q1. Further explanation on why the assumption of access to the optimal arm and the best reward is essential for solving the problem, how realistic is the relaxation described in Footnote 3, and any insights on whether these assumptions can be removed altogether in the future would be good to have.
>
> A1. Please see our answer to W2. The question regarding the necessity of these assumptions is a good one. If we ignore computational considerations, then one can simply search through subsets of the eliminated set and find a set of arms with low condition number. Note that we have proven that such a subset is guaranteed to exist, and this procedure, although inefficient, does not require knowledge of the optimal arm. Above, we have laid out a way to slightly alter our inverse learner such that we relax the assumption on the knowledge of $\mu^*$. We will have this ready for the Camera Ready Version.
>
> &nbsp;
>
> > Q2. A definition of "count number" (or a reference) should be given for those readers unfamiliar with this area.
>
> A2. To the best of our knowledge, we did not use this term in the main text - please let us know if we may have missed something. Do you mean phase number?
>
> &nbsp;
>
> > Q3. There are some minor typos, etc., which should be corrected in the final version.
>
> A3. We will certainly fix typos for the Camera-Ready Version. We would be happy to incorporate similar writing suggestions to improve readability for a broad audience.
>
> &nbsp;
>
> ## References
>
> Wenshuo Guo, Kumar Krishna Agrawal, Aditya Grover, Vidya Muthukumar, and Ashwin Pananjady. Learning from an exploring demonstrator: Optimal reward estimation for bandits. arXiv preprint arXiv:2106.14866, 2021.

---

### Official Review · Reviewer_8qMU · 2024-03-22

**Q2-1 Originality-Novelty:** 2
**Q2-2 Correctness-Technical Quality:** 3
**Q2-5 Clarity Of Writing:** 4

**Q1 Summary And Contributions:**

The authors consider inverse reinforcement learning applied to contextual multi-armed bandits, aiming at learning a policy from a learned agent with limited data. Their work is applied to Phased elimination algorithm and applied for several experiments in both synthetic and real-world data. Theoretical analysis is also provided.

**Q2-3 Extent To Which Claims Are Supported By Evidence:**

3: Good: the main claims are supported by convincing evidence (in the form of adequate experimental evaluation, proofs, (pseudo-)code, references, assumptions).

**Q2-4 Reproducibility:**

2: Fair: key resources (e.g. proofs, code, data) are unavailable but key details (e.g. proof sketches, experimental setup) are sufficiently well-described for an expert to confidently reproduce the main results.

**Q3 Main Strengths:**

The paper is well structured, clear and well written, it is also self-contained with all needed information for understanding. Experiments are quite extensive.

**Q4 Main Weakness:**

No major issue on this paper

**Q5 Detailed Comments To The Authors:**

The paper is very interesting and well written. There are 2 things I would have like to see to better highlight the benefits of the work, though not critical for publication :
-	The use of other MABs algorithms in experiments. I believe the propositions made in the paper could be extended to other algorithms instead of just Phased elimination, the paper would shine brighter by illustrating more applicability.
-	More real world dataset. It doesn’t seems critical since at least there is movieLens, synthetic data and theoretical proofs, however, while MovieLens seems to be a very rich dataset, it is not that much. According to numbers declared in paper, a user provide on average a rating for 156 arms, but there is a very big variability in it, most ratings being focused on a much smaller subsets of arms while many arms received few ratings. At the same time, ratings on this dataset tend to be very positive in most case. Hence, a random selection of users and arms may results in a quite small solution space with little risk for error. Another real-world dataset, even small but not concerned by those issues would further confirm the observation made.

**Q9 Complying With Reviewing Instructions:**

Yes

---

> ### Author Rebuttal · Authors · 2024-04-05
>
> We very much appreciate your thoughtful review and positive assessment of our paper. Below, we respond point-to-point to each of our concerns and suggestions. Please let us know if you have any follow-up questions, which we would be happy to answer.
>
> ## Weaknesses
>
> > W1. The use of other MABs algorithms in experiments. I believe the propositions made in the paper could be extended to other algorithms instead of just Phased elimination, the paper would shine brighter by illustrating more applicability.
>
> A1. We plan on exploring how to extend our insights to other algorithms, such as LinUCB, by using a constrained optimization approach to learn the reward vector (for example). However, due to the different structures in LinUCB, designing a new inverse learner and providing theoretical analysis is nontrivial and deserving of follow-up work.
> Having said, we believe some principles of our estimator could be generalized to LinUCB – by selecting arms that are a) suboptimal but are selected comparably often to the optimal arm $a^*$ (i.e., analogous to the set $\mathcal{A_L} \setminus \mathcal{A_{L-1}}$), and b) “well-conditioned” (analogous to Steps 3-7 in Algorithm 2). We believe our confidence-interval-based estimator and subsequent step of linear regression (Step 8 of Algorithm 2) could also be generalized. However, the details of analyzing the estimator from LinUCB in this way are likely to become much more complicated owing to the algorithm’s intricate day-to-day behavior.
>
> &nbsp;
>
> > W2. More real world dataset.
>
> A2. We are happy to consider running an additional experiment on an additional real-world dataset, such as the Amazon Music dataset, for the Camera-Ready Version. Specifically, just like we modelled user preferences for movies on the MovieLens dataset, we will learn user preferences on music based on their reviews.

---

### Official Review · Reviewer_bmcu · 2024-03-24

**Q2-1 Originality-Novelty:** 2
**Q2-2 Correctness-Technical Quality:** 3
**Q2-5 Clarity Of Writing:** 3

**Q1 Summary And Contributions:**

This paper introduces an inverse reinforcement learning method for stochastic linear bandits, focusing on learning from a single demonstration. It proposes a simple inverse learning procedure that estimates the linear reward function with a specified error bound. Key contributions include the development of an inverse estimator and its theoretical and empirical validation, including simulations and a semi-synthetic experiment with the MovieLens dataset.

**Q2-3 Extent To Which Claims Are Supported By Evidence:**

3: Good: the main claims are supported by convincing evidence (in the form of adequate experimental evaluation, proofs, (pseudo-)code, references, assumptions).

**Q2-4 Reproducibility:**

3: Good: key resources (e.g. proofs, code, data) are available and key details (e.g. proofs, experimental setup) are sufficiently well-described for competent researchers to confidently reproduce the main results.

**Q3 Main Strengths:**

This paper combines inverse reinforcement learning with stochastic linear bandits, tackling the problem with a novel single-demonstration approach. The idea is clear and well stated.

**Q4 Main Weakness:**

1. The method relies on specific strong conditions, such as the use of the Phased Elimination algorithm and assumptions about the action set's geometry.

2. The paper does not extensively compare its approach with alternative methods in stochastic linear bandits or explore how it might integrate with or improve upon existing IRL frameworks.

**Q5 Detailed Comments To The Authors:**

1. How does the proposed inverse learning procedure for stochastic linear bandits compare in efficiency and accuracy with traditional IRL methods that require multiple demonstrations, especially in high-dimensional action spaces?

2. Given the reliance on the Phased Elimination algorithm, how might the proposed method be adapted or extended to work with other forward algorithms in stochastic linear bandits?

3.How might the assumptions about the action set's geometry and density impact the method's performance in real-world scenarios? Is there a way to relax these assumptions while maintaining the method's effectiveness and theoretical guarantees?

**Q9 Complying With Reviewing Instructions:**

Yes

---

> ### Author Rebuttal · Authors · 2024-04-05
>
> We very much appreciate your thoughtful review and insights. Below, we respond point-to-point to each of our concerns and suggestions.
>
> ## Response to “Main Weaknesses”:
> W1. We agree that our results make assumptions, but believe this is a good starting point for studying the IRL framework with a single demonstration. We provide some context below.
>
> 1. The phased elimination algorithm is similar to the successive-arm-elimination (SAE) algorithm (Even-Dar et al., 2006) for multi-armed bandits, and we choose it as the forward algorithm as it is especially simple and interpretable. Elimination-based algorithms are similar to popular optimism-based algorithms; regret bounds are identical up to constant factors – the main difference manifests in their day-to-day behavior. Nevertheless, Guo et al., 2021 showed that similar inverse estimators could be designed for both algorithms for MAB. We similarly believe our estimation principles could be generalized to optimism-based linear bandit algorithms.
> 2. The first two parts of Assumption 4.1 require the action set to be sufficiently dense, i.e., a very large number of arms. This contrasts with the MAB setting, where the action set is very small or sparse. Linear bandit algorithms are especially motivated by large action set settings, where we gain significantly from sharing information across arms. The third part of Assumption 4.1 concerns the shape of the action set and we believe it may be necessary –  without this, we would need to handle "discontinuous" action sets for which there is a spike in reward near the optimal arm. It is fundamentally challenging to perform inverse estimation in such cases.
>
> Even under our assumptions, the problem of inverse learning is much more challenging than in the corresponding MAB case.
>
> &nbsp;
>
> W2. Thank you for this suggestion. Please refer to our answer to Q1 for comparison and integration with existing IRL frameworks. We emphasize two points within linear bandits:
> 1. Our work is the first to establish that consistent inverse estimation is possible from a single demonstration of linear bandit. Our estimator and analysis require new ideas compared to the simpler MAB case.
> 2. Existing IRL frameworks that require multiple demonstrations could be applied to the stochastic linear bandit setting. The earliest IRL framework that assumed optimal $a^*$ demonstration would cause identifiability issues. The max-ent framework would work but be computationally inefficient due to the combinatorial size of the action set in real-world applications of linear bandits. In contrast, linear bandit trajectories enable learning even from a single demonstration and are more efficient to implement than max-ent (refer to our response to Reviewer gmoF, W3 for more details on the complexity of both forward and inverse algorithms).
>
> &nbsp;
>
> ## Response to “Detailed comments to the authors”:
>
> Q1. Our method can achieve consistent estimation (in T) with only a single demonstration, while traditional IRL methods (such as Max-Ent) achieve consistent estimation in the number of demonstrations $n$ (at a rate of $O(1/n)$ in squared error). Since these are different settings, they are not directly comparable in terms of sample efficiency. For computational efficiency, learning from a single bandit demonstration should be more efficient than learning from multiple Max-Ent demonstrations; see our response to W2. The principle of Max-Ent is to provide an unbiased estimator of the reward from a single demonstration. Thus, averaging over n demonstrations gives error $O(1/n)$ by reducing variance. This suggests that we could adapt our estimator to n bandit trajectories each of length $T$ thus naturally integrating our approach with many demonstrations. We conjecture that the rate of Theorem 4.1 would be multiplied by $1/n$. This is a strictly better rate than the Max-Ent estimator because of the additional decay in $T$ arising from Theorem 4.1.
>
> &nbsp;
>
> Q2. Some principles of our estimator could be generalized to LinUCB – by selecting arms that are a) suboptimal but are selected comparably often to the optimal arm $a^*$ (i.e., analogous to the set $\mathcal{A_L} \setminus \mathcal{A_{L-1}}$), and b) "well-conditioned" (analogous to Steps 3-7 in Algorithm 2). We believe our subsequent step of linear regression (Step 8 of Algorithm 2) could also be generalized. However, the details will become much more complicated owing to LinUCB’s intricate day-to-day behavior.
>
> &nbsp;
>
> Q3. Empirically, our assumptions are well-founded – Section 6.1 demonstrates that inverse estimation is consistent for action sets that are $\ell_p$-balls, and our MovieLens experiments in Section 6.2 demonstrate that our algorithm performs well on real-world action set shapes. Theoretically, we believe these assumptions are needed – please also refer to our answer to W1 above.
>
> &nbsp;

---

### Official Review · Reviewer_gmoF · 2024-03-26

**Q2-1 Originality-Novelty:** 2
**Q2-2 Correctness-Technical Quality:** 2
**Q2-5 Clarity Of Writing:** 2

**Q1 Summary And Contributions:**

The authors study the problem of learning from a single sequence of actions of a stochastic linear bandit algorithm. They propose an inverse reinforcement learning algorithm for doing this and analyze its regret. Furthermore, simulated experiments are provided.

**Q2-3 Extent To Which Claims Are Supported By Evidence:**

2: Fair: the main claims are somewhat supported by evidence (but the experimental evaluation may be weak, or does not match entirely with the claims, important baselines may be missing, proofs contain important ideas but lack rigor, algorithmic details are only discussed superficially, references are imprecise, assumptions are not sufficiently motivated or explicated, etc.).

**Q2-4 Reproducibility:**

2: Fair: key resources (e.g. proofs, code, data) are unavailable but key details (e.g. proof sketches, experimental setup) are sufficiently well-described for an expert to confidently reproduce the main results.

**Q3 Main Strengths:**

- The problem is interesting and relevant within the bandits field.
- The theoretical results are noteworthy.
- The assumptions are fairly mild.
- The readability is good.

**Q4 Main Weakness:**

- The theoretical analysis could be made more thorough and clear.
- The experiments using small test examples are helpful. However, it would make the experiments richer if some real-world application(s) as pointed out in the introduction of the paper were used for the experiments.
- Runtime and space analysis of the algorithms should be done.

**Q5 Detailed Comments To The Authors:**

Please refer to the weaknesses above.

**Q9 Complying With Reviewing Instructions:**

Yes

---

> ### Author Rebuttal · Authors · 2024-04-05
>
> We very much appreciate your thoughtful review and insights. Below, we respond point-to-point to each of our concerns and suggestions. We appreciate your openness to increasing your score and hope our response can clarify and address your concerns. Please let us know if you have any follow-up questions, which we would be happy to answer.
>
> ## Reproducibility
>
> > R1. Key resources (e.g., proofs, code, data) are unavailable.
>
> A1. Please note that full proofs are provided in Appendix A-F and the pseudocode used for our implementation is provided in Appendix G. The dataset we use is MovieLens 25M, which is open-source and publicly available. Our code and data are available in 'InvPhasedElim.zip,' which we submitted as part of the supplementary material. We would be happy to make references to these sections clearer in the main paper so that readers do not miss these resources.
>
> &nbsp;
>
> ## Weaknesses
>
> > W1. The theoretical analysis could be made more thorough and clear.
>
> A1. We want to highlight that the proofs for all the theorems and technical lemmas are available in the Appendix (please see `ALT_2024_Submission (2)-11-24.pdf`). We welcome any suggestions you have to improve the clarity of exposition.
>
> &nbsp;
>
> > W2. The experiments using small test examples are helpful. However, it would make the experiments richer if some real-world application(s) as pointed out in the introduction of the paper were used for the experiments.
>
> A2. In Section 6.2, we use our Inverse Learning method in the real-world application of recommender systems (we will also reference this in the introduction), particularly using the MovieLens 25m dataset to predict users' movie tastes based on their previous movie ratings. While this is ultimately a simulated environment, the parameters of the simulation are learned from real data. We hope this convincingly demonstrates the practicality of our algorithm in real-world scenarios, and we believe this template could also be applied to other real-world applications.
>
> &nbsp;
>
> > W3. Runtime and space analysis of the algorithms should be done.
>
> A3. Thanks for the good suggestion; we will discuss runtime and space analysis of the algorithms in the camera-ready version. The forward algorithm, Algorithm 1 (phased-elimination), is a classical algorithm whose runtime was analyzed in prior work [Valko et al, '14; Lattimore and Szepesvari, '20]; the time complexity depends greatly on your implementation of finding the G-optimal design. However, the runtime of the forward algorithm is not especially relevant to our analysis as the trajectory of the forward algorithm has already been given to us.
> More relevant and important is the runtime and space analysis of our inverse estimator, Algorithm 2. We describe this analysis here and will include it in the camera-ready version. Note that Step 8 is a linear least-squares problem that can be solved in $O(d^3)$ time or less. Steps 1-2 are initialization steps which are constant-time. Steps 3-7 construct the linearly independent arms $a^1,\ldots,a^d$, which proceeds in a sequence of steps: a) constructing the equiangular tight frame $s_1,\ldots,s_d$, b) rotating $a^*$ in the plane spanned by $a^*$ and $s_i$ by a small angle $\beta$ to construct each of the arms $a^1,\ldots,a^d$. Step a) can be done in $\mathcal{O}(d^2)$ time for each arm, leading to a $\mathcal{O}(d^3)$ complexity in total as there are $d$ arms. Similarly, step b) can be done in $\mathcal{O}(d^2)$ time for each arm (and could be even faster depending on the structure of the rotation). Therefore, Algorithm 2 can be run efficiently in $\mathcal{O}(d^3)$ time. From the point of view of memory, Algorithm 2 only needs to create additional memory for the linearly independent arms $a^1,\ldots,a^d$ and their corresponding estimated rewards. This creates a memory requirement of only $\mathcal{O}(d^2)$ (over and above the given trajectory of the forward algorithm).
>
> &nbsp;
>
> ## References
>
> Michal Valko, Remi Munos, Branislav Kveton, and Tomáš
> Kocák. Spectral bandits for smooth graph functions. In
> Eric P. Xing and Tony Jebara, editors, Proceedings of
> the 31st International Conference on Machine Learning,
> volume 32 of Proceedings of Machine Learning Research,
> pages 46–54, Bejing, China, 22–24 Jun 2014. PMLR.
> URL https://proceedings.mlr.press/v32/
> Valko14.html.
>
> Tor Lattimore and Csaba Szepesvári. Bandit Algorithms. Cambridge University Press, 2020. doi: 10.1017/ 9781108571401.

---

### Meta-Review · Area_Chair_bfEa · 2024-04-20

This submission introduces a novel inverse reinforcement learning (IRL) technique for stochastic linear bandits that operates effectively with a single demonstration. This approach is especially significant as it addresses the challenge of learning reward functions efficiently in environments with limited data. The authors propose a method that estimates the linear reward function consistently, underpinned by a detailed theoretical analysis and empirical validation through simulations and semi-synthetic experiments using the MovieLens dataset.

A significant concern among reviewers is the paper's reliance on specific algorithms (Phased Elimination) and assumptions about the action set's geometry. However, the authors provided a thorough justification for these choices and discussed potential generalizations in their rebuttal, which seemed to satisfy most concerns.